# Genetics, Functions, and Clinical Impact of Presenilin-1 (PSEN1) Gene

**DOI:** 10.3390/ijms231810970

**Published:** 2022-09-19

**Authors:** Jaya Bagaria, Eva Bagyinszky, Seong Soo A. An

**Affiliations:** 1Department of Bionano Technology, Gachon University, Seongnam 13120, Korea; 2Department of Industrial and Environmental Engineering, Graduate School of Environment, Gachon University, Seongnam 13120, Korea

**Keywords:** early onset Alzheimer’s disease, presenilin-1, mutation, ACMG-AMP guidelines, γ-secretase

## Abstract

Presenilin-1 (PSEN1) has been verified as an important causative factor for early onset Alzheimer’s disease (EOAD). PSEN1 is a part of γ-secretase, and in addition to amyloid precursor protein (APP) cleavage, it can also affect other processes, such as Notch signaling, β-cadherin processing, and calcium metabolism. Several motifs and residues have been identified in PSEN1, which may play a significant role in γ-secretase mechanisms, such as the WNF, GxGD, and PALP motifs. More than 300 mutations have been described in PSEN1; however, the clinical phenotypes related to these mutations may be diverse. In addition to classical EOAD, patients with PSEN1 mutations regularly present with atypical phenotypic symptoms, such as spasticity, seizures, and visual impairment. In vivo and in vitro studies were performed to verify the effect of PSEN1 mutations on EOAD. The pathogenic nature of PSEN1 mutations can be categorized according to the ACMG-AMP guidelines; however, some mutations could not be categorized because they were detected only in a single case, and their presence could not be confirmed in family members. Genetic modifiers, therefore, may play a critical role in the age of disease onset and clinical phenotypes of PSEN1 mutations. This review introduces the role of PSEN1 in γ-secretase, the clinical phenotypes related to its mutations, and possible significant residues of the protein.

## 1. Introduction

Neurodegenerative dementia is classified as a major health issue, with more than 50 million people around the world affected by some form of dementia. Most affected patients (almost two-thirds) are diagnosed with Alzheimer’s disease (AD), followed by frontotemporal dementia (FTD), dementia with Lewy bodies (DLB), and vascular dementia [1]. AD is an irreversible and progressive form of dementia associated with the loss of memory and cognitive function. Age has been verified as the strongest risk factor for AD, as disease prevalence increases with age. The number of patients with AD increases rapidly after 65–70 years of age [2]. AD has different neuropathological hallmarks, including extracellular amyloid plaques, intracellular neurofibrillary tangles (NFTs), and the loss of neurons and synapses. Additional atypical neuropathological phenotypes, such as cerebral amyloid angiopathy (CAA), cotton wool plaques, Lewy bodies or Pick’s bodies may also co-occur in patients with AD [3]. Although the majority of cases are associated with the late-onset form of the disease, several cases develop a disease phenotype at a younger age. Early-onset AD (EOAD) represents 1–5% of all AD cases and can occur under 65 years of age. The majority of patients present with disease phenotypes in their 40s or 50s, but disease onset at a young age (in their 20s and 30s) is also possible [4]. EOAD is usually characterized by an autosomal dominant inheritance pattern; however, autosomal recessive inheritance may also be possible [5]. Three genes have been identified as causative factors for EOAD: amyloid precursor protein (APP) on chromosome 21, presenilin-1 (PSEN1) on chromosome 14, and presenilin-2 (PSEN2) on chromosome 1 [4]. These mutations are rare, as only 1% of patients carry APP mutations, 6% of patients carry PSEN1 mutations, and less than 1% of patients carry PSEN2 mutations [4]. Mutations in PSEN1 have been identified as the most common causative factor for EOAD, and patients with these mutations may present with a rapidly progressive disease [4,6]. To date, more than 300 mutations have been identified in PSEN1 [https://www.alzforum.org/mutations/psen-1, accessed on 1 July 2022]. In this review, we discuss the functions of PSEN1 in Alzheimer’s disease and its possible impact on other diseases. Additionally, we introduce PSEN1 mutations with their clinical phenotypes and the functional studies performed on these mutations.

### 1.1. PSEN1 Structure and Functions

The PSEN1 protein contains 467 amino acids and shows 65% homology to the PSEN2 protein. PSEN1 is a transmembrane protein with nine transmembrane domains connected to hydrophilic loops in either the extracellular area or the cytosol. The N-terminal fragment and the large hydrophilic loop are located in the cytosol region, while the C-terminal fragment is located in the extracellular space. The large hydrophilic loop also contains a membrane-associated area. PSEN1 is a member of the γ-secretase complex that plays a key role in APP processing and amyloid peptide generation [7]. PSEN1 contains two catalytic aspartic acids (Asp275 and Asp 385, located on TM6 and TM7, respectively), that play key roles in both PSEN1 endoproteolytic activity and γ-secretase activity [8]. Additional critical residues were found in TM, located between positions 95 and 98. Mutations in these residues can inhibit or abolish PSEN1 proteolysis and γ-secretase activity [9]. An additional important residue could be Asp345, which is located in the loop between TM6 and TM7. A caspase cleavage site was located next to this amino acid, which plays a crucial role in the interaction between PSEN1 and its binding partners [8]. As part of the γ-secretase complex, PSEN1 or PSEN2 forms a complex with nicastrin (Nct), PSEN enhancer 2 (PEN2), and anterior pharynx 1 (APH-1). Furthermore, the γ-secretase complex is known to interact with various other proteins, including ubiquitin, protein-folding related proteins or adhesion molecules [10]. PSEN1 is involved in the C-terminal transmembrane region of APP and the production of amyloid-β peptide (Aβ42) by γ-secretase, after APP processing α-and β-secretases. PSEN1 may not be an enzyme itself, but a crucial regulator protein in γ-secretase cleavage. PSEN1 could also play a role in the transport of C-terminal APP fragment to the gamma secretase complex. A deficiency in PSEN1 function may impair APP processing, resulting in altered amyloid peptide production [11]. Several presenilin domains may be important for protein functions. The C-terminal fragment interacts with Nct protein. Additionally, it can control γ-secretase activity through ATP binding via its nucleotide binding site. Furthermore, the C-terminal fragment interacts with TM1 and binds to APP [12]. The N-terminal region of PSEN1 (TM1, HL1, and TM2) plays a crucial role in the catalysis of substrates of the γ-secretase complex. Additionally, these domains may impact PSEN1 endoproteolysis and the coordination of substrate docking to the enzyme [13]. TM2 and TM3 interaction may impact the conformation of γ-secretase (either “semi-open” or “fully open”), and their interaction could control the availability of the enzyme’s active site. The “semi-open” conformation of the enzyme could be ideal for the production of short amyloid, while the mutations may induce the “open” conformation of the enzyme and longer amyloid peptide production [14]. TM5 and TM6 may be responsible for endoproteolysis and the maturation of the PSEN1 protein by controlling the “gate-plug mechanism” [15]. The TM4 domain of PSEN1 directly interacts with PEN2, enhancing the endoproteolysis of PSEN2 and the impact of γ-secretase activation [16]. The N-terminal fragment of PSEN1 and the large loop between TM6 and TM7 may not be essential for amyloid cleavage. The TM6 and TM7 domains contain two catalytic aspartates [17]. A water-containing cavity was identified inside the membrane between TM6 and TM7, enabling hydrolysis inside the lipid bilayer [12]. In the following section, the roles of PSEN1 domains and probable significant residues are introduced in detail.

Along with the APP, the γ-secretase complex can splice several additional substrates; PSEN1 (and PSEN2) can also impact Notch signaling through the cleavage of Notch receptors [18]. Notch signaling has been verified to determine cellular fate and control cell differentiation and division. PSEN1 is essential for cleavage of Notch receptors within their TM domains. PSEN1 controls the release of the Notch receptor intracellular domain (NICD) and allows its entry into the nucleus [19]. PSEN1 may also be involved in Wnt signaling by controlling β-catenin stability. PSEN1 can promote the phosphorylation of β-catenin and inhibit cyclin D1, CDK6, and c-Myc molecules, as well as cell-cycle progression [20,21]. PSEN1 mutations may impair Wnt signaling either directly or indirectly. PSEN1-mediated amyloid-β toxicity may result in downregulation of Wnt signaling and neuronal death. Contrastingly, PSEN1 mutations could possibly upregulate Wnt signaling, leading to abnormal cell cycle events and neuronal loss [21,22]. PSEN1 regulates epidermal growth factor receptor (EGFR) trafficking and turnover. PSEN1 loss or dysfunction has been suggested to reduce EFGR1 turnover. This process impairs EFGR trafficking from endosomes to lysosomes [23]. PSEN1 may impact phospholipase C (PLC) and protein kinase C (PKC) activation. In terms of PSEN1 (and PSEN2) knockout, the expression of most PKC and PLC isoforms was reduced. However, PKC delta levels were higher in PSEN1 knockout cells, resulting in a higher degree of apoptosis and reduced mitochondrial membrane potential. Through different PLC and PKC isoforms, PSEN1 may be involved in the regulation of calcium signaling [24]. Through calcium homeostasis, PSEN1 (and PSEN2) may affect synaptic homeostasis. PSEN1 deficiency may be related to impaired calcium transport and synaptic dysfunctions [25]. Taken together, PSEN1 may play a role in multiple molecular pathways (Figure 1).

PSEN1 and PSEN2 share 66% homology in their sequences, and both are members of γ-secretase. However, PSEN1 seems to have a stronger impact on AD progression than PSEN2. Through Alzforum, PSEN1 and PSEN2 mutations are known to occur in 300 and 87 mutations, respectively. Most studies seem to focus on PSEN1, while simply overlooking PSEN2 [26]. Another explanation could be that PSEN1 and PSEN2 may be involved in different aspects of the γ-secretase process. Expression levels of PSEN1 were higher in both mouse and human brains during development, compared to those of PSEN2 [27,28]. Even though both proteins have endopeptidase and carboxypeptidase activities, PSEN2-type γ-secretase complexes were revealed to have lower levels of both activities, compared to those of the PSEN1 type. Mouse models revealed that germline PSEN1 knockout may result in serious dysfunctions; for example, it may disturb skeletal and brain developments [29]. Meanwhile, knockout for PSEN2 may also result in dysfunctions, but less aggressive ones [30]. The γ-secretase functions seemed to be different depending on whether it assembles with PSEN1 or PSEN2 [31]. A study by Yonemura et al. examined the γ-secretase activity of PSEN1 and PSEN2 in yeast models. PSEN1-related γ-secretase presented significantly higher activity compared to PSEN2 related γ-secretase in the β-galactosidase assay. Amyloid peptide production resulted in higher amyloid production in the case of the PSEN1-related γ-secretase complex. However, this study also compared the PSEN1 and PSEN2 activity per one γ-secretase complex, and suggested that PSEN1 and PSEN2 may have a similar degree of activity. Co-immunoprecipitation of PSEN1 and PSEN2 with other γ-secretase proteins revealed that PSEN2 concentration was lower in the complex, compared to PSEN1, suggesting that PSEN1 and PSEN2 may have different affinity to the other γ-secretase component proteins [32]. ThePSEN1-γ-secretase complex presented significantly higher activity in comparison to PSEN2-related γ-secretase in the β-galactosidase assay. The amyloid peptide production was also higher in the case of the PSEN1-related γ-secretase complex [33]. However, no significant differences were detected in the production of the APP intracellular domain (AICD) and Notch intracellular domain (NICD) [33]. Watanabe et al. (2021) analyzed the PSEN1 and PSEN2 functions in induced pluripotent stem cells (iPSC) cell lines. This study revealed that the PSEN1-related γ-secretase complex was able to cleave the N-cadherin, but not the PSEN2-related γ-secretase complex. Both defective PSEN1- and PSEN2-related γ-secretase may impair amyloid cleavage. The distribution of PSEN1- and PSEN2-related γ-secretase may also be different. PSEN2 may direct the γ-secretase process toward the late endosome-lysosome system. Meanwhile, PSEN1-related γ-secretase could be distributed more broadly inside the cell. The different localizations of PSEN1 or PSEN2 may result in different mechanisms of substrate processing of γ-secretase [31]. Taken together, additional studies are needed to establish why PSEN1 mutations more commonly impact EOAD, but not PSEN2. Either PSEN2 is less extensively studied than PSEN1 or PSEN1 may have higher impact on APP cleavage compared to PSEN2.

### 1.2. PSEN1 Mutations and Their Classification

As mentioned before, PSEN1 contains nine transmembrane domains with hydrophilic loops, including a long loop between TM6 and TM7. These PSEN1 domains can catalyze the cleavage of γ-secretase substrates, including APP and Notch proteins. To date, more than 300 mutations have been described in PSEN1 (Figure 2, Appendix A). The majority of these have been associated with EOAD, but they may also be connected with other disease phenotypes, such as frontotemporal dementia (FTD) or Parkinson’s disease (PD) [19,34]. New sequencing techniques, such as next-generation sequencing (NGS), whole-genome sequencing (WGS) and whole-exome sequencing (WES) reduce the time and cost of genetic analysis and accelerate the discovery of novel, probably disease-related mutations [35].

The pathogenic nature of PSEN1 (and PSEN2) mutations may be categorized into four categories, as established by Guerreiro et al. (2010): not pathogenic (or risk variants), possibly pathogenic, probably pathogenic, or definitely pathogenic. This algorithm suggests several aspects that may predict the impact of a mutation on the disease. First, the mutation should segregate with the disease. If the mutation is found in several affected family members (but not in the unaffected), it leads to strong evidence that the mutation is pathogenic. Second, mutations should not appear in the control group. If the mutation appears in disease cases, but not in more than 100 controls, it may also be probably or possibly pathogenic. If the mutation was found in the controls, the pathogenic nature of the mutation may be refuted. In addition, the residue properties may also be important. Therefore, if a mutation affects a conserved residue between PSEN1 and PSEN2, there is a strong chance of it being pathogenic. It may also be important to determine whether any pathogenic mutations were found in the affected residue or if the mutation followed the helix rule. Finally, mutations should be associated with a higher Aβ42/40 ratio compared to controls [36]. The most recent analyses re-evaluated the pathogenic nature of EOAD-associated mutations following the guidelines of the American College of Medical Genetics and Genomics and the Association for Molecular Pathology (ACMG-AMP). This algorithm categorizes mutations as pathogenic, likely pathogenic, benign, likely benign, or variant with uncertain significance (VUS). The first study of the ACMG-AMP guidelines was introduced by Richard et al. in 2015 for all Mendelian diseases [37]. They designed an evidence framework based on different criteria including population data, in silico analyses, functional studies, and segregation data. This study suggests that variants may be null mutations, frameshifts, START codon-affecting variants, splice site mutations, stopgain, or stoploss variants, and could be treated as potential strongly pathogenic variants. Furthermore, positive family history, additional mutations in the same residue, functional studies, and higher prevalence in patients than in controls also suggest as strong proof that the mutation may be pathogenic. Additional strong proof could be that the mutation may occur in more than one independent disease case. Moderate proof of the pathogenic nature of the mutation could be its absence or very low frequency in reference databases (gnomAD; https://gnomad.broadinstitute.org or 1000Genome; https://www.internationalgenome.org, all accessed on 1 July 2022). Furthermore, variants may affect a mutation hotspot, or a functional or conserved/critical residue. In silico tools may also be useful for prediction; for example, variants with CADD (https://cadd.gs.washington.edu/snv, accessed on 1 July 2022) scores over 20 were predicted to be damaging. Possible benign variants may appear at a higher frequency among the control in comparison to that in the patients with the disease. Functional and in silico studies did not reveal any damaging effects from the benign variants. Furthermore, putative benign variants may not segregate with the disease [37,38,39]. Table 1 summarized ACMG-AMP classification criteria for the PSEN1 (APP, PSEN2) mutations. Among the PSEN1 mutations included in Alzforum (https://www.alzforum.org/mutations/psen-1, accessed on 1 July 2022), six missense mutations were categorized as benign or likely benign. Several mutations could not be classified by Alzforum, as the available data on them may be limited. These mutations were found in a single patient. Furthermore, no segregation could be made since no data were available on family members or relatives of proband patients refused the genetic test. The majority of PSEN1 variants in Alzforum were categorized as pathogenic or likely pathogenic, as mutations fulfilled several criteria of the ACMG-AMP guidelines for pathogenic variants [38,39].

### 1.3. PSEN1 Mutations and Significant Residues at Each Domain

Twelve mutations were found in the N-terminal fragment, although their pathogenic natures remain unclear. Two mutations, Asp40del (delGAC) and Ala79Val, were suggested to have the strongest impact on the disease onset [39,40,41,42]. Asp40del has been identified in individuals with AD and FTD. The deletion (delGAC) was introduced into mouse neuroblastoma cells, and the mutation was suggested to increase amyloid-β (Aβ) 42 production, but it did not impact the production of Aβ40 [39]. Furthermore, this variant was also associated with reduced levels of long amyloid peptides (Aβ43, Aβ42) in the cerebrospinal fluid (CSF); however, it may not affect the level of short amyloid peptide (Aβ40) [41]. Ala79Val has been observed in several cases in Europe and the USA, and the disease phenotypes may be variable. EOAD with a positive family history is prominent in several cases [42,43,44]. PSEN1 Ala79Val is also associated with late-onset AD (LOAD), either with a positive or a negative family history [45]. Furthermore, other disease phenotypes may also be possible among mutation carriers, including PD [46], FTD [47] or vascular AD [41]. Functional studies have revealed that mutations appearing in GnomAD may be significant in disease onset. Carriers of mutations are associated with reduced amyloid levels in the CSF. In HEK293 cells, Ala79Val reduced Aβ40 levels, resulting in a higher Aβ42/40 ratio [48]. In mouse fibroblasts with this mutation, Aβ42 levels were elevated, but Aβ40 levels and total Aβ levels were not changed [44]. The remaining variants were suggested to be benign variants or variants with unclear significance (VUS), even though they may impact the disease risk. The majority of these mutations, including Asn24Ser [49], Arg35Glu [50,51] or Arg42Leu [52], were observed in EOAD patients, but other clinical hallmarks were also observed, including FTD in Glu15His [52], Pick’s disease in case of Pro49Leu [41] or PD with non-amnestic MCI in case of Arg41Ser [53]. It has been suggested that the N-terminal fragment may not play a significant role in amyloid peptide generation; however, the presence of probable pathogenic mutations in this region suggests that it may have some impact on disease mechanisms [16].

Several mutations appeared in the TM1 region of PSEN1, the majority of which were confirmed to affect AD. The first TM domain contains several residues that could greatly impact the γ-secretase cleavage. Residues in TM1 are highly conserved [54]. Furthermore, TM1 is located in proximity to critical motifs of γ-secretase, such as GxGD and PALP, in TM7 and TM9, respectively. Mutations in TM1 may result in stress during the interactions between TM1 and TM7 (or TM9). Mutations in TM1 may also increase the distance between these helices [13,55,56,57,58,59]. Several residues were found to reduce or even completely abolish the γ-secretase activity, or impact other mechanisms (Figure 3). The N-terminal domain of TM1 (residues between 87–90) was suggested to play a role in the γ-secretase process, while the C-terminal motif of TM1 (residues between 95 and 98) could impact both γ-secretase and presenilinase activity. Val96 has been suggested to play an extremely critical role in the binding of APP and Notch proteins. In addition to the elevated Aβ42/40 ratio, mutant Val96 may also impair the autoproteolytic endopeptidase activity of PSEN1 [9]. Cryo-electron microscopy analysis revealed that Leu85 may impact the APP binding [60]. Mutant Pro88 was suggested to impair the secretion of apolipoprotein E (APOE), since the Pro88Leu mutation was associated with reduced blood APOE levels [61]. The majority of TM1 mutations were associated with disease onset in patients in their 40s or 50s; however, disease onset at a young age was also prominent in several cases. Patients with Leu85Pro [62,63] or Pro88Leu developed their first disease symptoms in their 20s [64,65,66,67,68,69]. Additional mutations were associated with disease onset at a young age; for example, Val89Leu with G>C exchange [64] or Ile83_Met84del [65] or Val97Leu [66] carriers may develop disease phenotypes in their 30s. Most patients with the above mutations developed EOAD, but other symptoms may also be prominent. For example, patients with Ile83_Met84del developed spastic paraparesis, and neurological findings revealed cotton wool plaques, mostly present in the neocortex, hippocampus, or striatum [65]. One of patients with Met84Val demonstrated spastic paraparesis with ideomotor apraxia as the disease progressed [67,69]. Behavioral changes (depression, irritability, aggression, anxiety, or hallucinations) appeared in different mutations, including Ile83Thr [68], Met84Val [69], Val89Leu (G>T) [59], Val96Phe [54]. PSEN1 Leu85Pro was associated with an atypical form of AD in both patients. Both cases reported disease onset at a very young age, and the disease course was aggressive. A Japanese patient presented with spastic paraparesis alongside visual dysfunctions (visuospatial agnosia, simultaneous agnosia, and optic ataxia), suggesting that this case was related to a visual variant of AD [62]. The second patient with Leu85Pro developed corticobasal syndrome. Although she initially presented with memory and mood impairments, she also developed Parkinsonism and myoclonus [63]. The patient with Pro88Leu initially developed myoclonus in her 20s, followed by memory decline at 41 years of age. Additional atypical features, including Parkinsonism, ataxia, spasticity, and dystonia, also appeared in the patient [64]. The patients with Cys92Ser were diagnosed with atypical EOAD. In addition to memory decline, patients presented several atypical Parkinson’s-like features, including extrapyramidal signs, rigidity, and bradykinesia. Psychiatric symptoms, hallucinations, and delusions may also be present [70]. One member of a Tunisian family with the Ile83Thr [68,71] mutation developed refractory epilepsy. At the age of 32, the patient developed deficits in attention and memory as well as depression. Although epilepsy may be associated with different PSEN1 mutations, this case presented an atypical early-onset epilepsy. Additionally, other family members with Ile83Thr mutation did not present epilepsy. Further studies are required to elucidate this mutation and its relationship with epilepsy [71].

Several pathogenic mutations were found in the HL1 loop, located between TM1 and TM2, and this region was verified as a hot spot for pathogenic mutations involved in EOAD [72]. Among them, 28 mutations were categorized as missense and three as indels in this region. HL1 was verified to play a critical role in the stepwise processing of γ- secretase. Furthermore, HL1 can also impact the γ-secretase modulation. Several residues were identified to be critical in γ-secretase processing (Figure 4). For example, Phe105, Leu113, Tyr115, Thr116, and Pro117 impact the carboxypeptidase activity, which can result in the cleavage of shorter amyloids. At least three pathogenic mutations were identified in these residues, which were verified to impact the longer amyloid peptide production [73,74,75] Furthermore, Glu120 may also be a significant residue in HL1, since four variants were described for this amino acid. Mutant Glu120 in iPSC cells was associated with increased phosphorylated Tau levels and enhanced mitochondrial dysfunctions [73,76]. The majority of these mutations were related to disease phenotype, resulting in typical EOAD presentation [72]. However, other neurodegenerative disease phenotypes were also related to PSEN1 HL1 mutations. For example, Leu113Pro, Thr122Ala, and Ser132Ala were found in patients diagnosed with FTD [52]. Furthermore, Ser132Ala was also found in a family presenting with DLB [77]. Arg108Pro was observed in a patient with cognitive degermation and myoclonic jerks [78]. Leu113Gln was associated with early-onset dementia, spasticity, and myoclonic jerks. Several mutations were related to EOAD and behavioral impairment (for example, depression, mood swings, apathy, stereotyped behavior, abulia, or delusion), including Pro117Leu [79,80], Pro117Ser [81], Thr119Ile [82,83,84], Glu120Lys [85] and Glu123Lys [85]. Four mutations resulted in language impairment, including Thr116_117delSer, Thr [86], Thr119Ile [83,84], Glu120Lys [84]. Furthermore, mutations such as Leu113indel [87], Tyr115His [77], Thr116indel [86], Pro117Ala [88] Pro117Leu [79], Pro117Ala [88] or Glu120Lys [84] were associated with motor impairments, such as myoclonic jerks, ataxia, gait impairment. Mutations in residue 113 [89], residue 117 [88], Glu120Asp [90] and Glu120Gly [91] and His131Arg [92] were associated with epilepsy and seizures. Two mutations, Thr116indel [87] and His131Arg [92] were also associated with headaches. Cotton wool plaques seem to be uncommon with mutations in HL1, since only one family with Thr116Asn presented cotton wool plaques [93].

In the second TM domain, 32 mutations appeared, all categorized as missense mutations. In the second HL loop, three missense mutations and one indel were found. Three mutations were described with at least two different codon combinations—Met139Ile, Met146Ile, and Met146Leu. Similar to TM1 and HL1, TM2 may have a strong impact on γ-secretase cleavage. Deletion in the region of the first two TM domains (TM1, HL1, and TM2) may result in abnormal activity of γ-secretase. Several residues in TM2 may impact the endoproteolytic process of PSEN1 protein. Furthermore, these residues may be involved in regulating the interaction of substrate docking site and the γ-secretase catalytic core (Figure 5) [13]. Several residues, including Met139, Ile143, and Met146, suggested to be located near the APP helix, may impact the formation of substrate binding core. Met146 and Thr147 may have close contact with the APP helix. Val151 and Tyr159 may play a significant role in the stabilization of APP and PSEN1 hybrid through a β-sheet formation and thus initiate the γ-secretase cleavage [60]. Tyr154 mutations may result in dysfunctions of endopeptidase activity [94]. Besides γ-secretase impairments, additional dysfunctions were also related to mutations in Ala136, Met139, and Met146. Mutations in residue 139 could have mild effect on endoproteolytic cleavage of Notch3 [95]. Variants in Ala136 (Ala136Gly) and Met146 (particularly, Met146Leu) could also be involved in calcium metabolism by accelerating cleaving of the calcium sensor STIM1, resulting in impaired calcium influx [92,96]. Furthermore, Met146 mutations (Met146Ile) could impair the autophagy and lysosomal function, resulting in abnormal lysosomal proteolysis and autophagosome clearance [97] A third mutation in Met146, Met146Val, could result in reduced neuroprotection by altering the trophic factor functions, for example BDNF or eB1 [98]. The mutation could also alter cerebral blood flow and reduce new blood vessel formation from the cleavage of ephrinB2 by the γ-secretase [99]. Disease onset at a young age may be commonly associated with mutations found in TM2, and in addition to the typical AD phenotypes, these mutations may result in atypical symptoms as well, such as motor-, personality-, or visual impairments [4]. Four mutations were associated with FTD or an FTD-like disease phenotype: Ala137Thr [52], Met139Val [100] and Met146Val [92,101] and Met146Leu [97]. Personality changes and language impairment were common among patients with different TM2 mutations, for example, Asn135Ser [98], Met139Val [101], Val142 [102,103], and Met146Ile [104]. Patients with Thr154Asn were initially diagnosed with spastic paraparesis in their 30s, and they developed AD phenotypes in their 40s [105]. Motor symptoms were also frequently associated with mutations in TM2. Furthermore, ten mutations, including mutations in Asn135 [89,106,107], Met139Val [100,108], Leu153Val [109], Ile143 [110,111,112] or Met 146 [113] were associated with myoclonus or myoclonic jerks. Spasticity was found to be associated with six TM2 mutations, including Asn135 [89,107,109], Met139Val [114], Thr147Ile [115], and Tyr154Asn [116]. Tyr159Cys was associated with Parkinsonism [84]. Seizures or epilepsy also frequently occurred among patients with TM2 mutations, for example, Asn135Ser [89], Met139Thr [117], Met139Val [108], Met143Ile [96], or Leu153Val [118]. Visuospatial impairment was associated with four mutations, including Met139Leu [119], Met139Val [101], Tyr159Cys [84], and Tyr159Pro [120]. Lewy bodies may not be common among patients with TM2 mutations, since they were only associated with Leu153Val mutation [107,109,118,121]. Additional rare symptoms included akinetic mutism and hallucinations, which were found in patients with Met139Val [101,108] and Ile143Val, respectively [122].

In TM3, to date, 44 pathogenic or probable pathogenic mutations have been reported, suggesting that this region may be critical in γ-secretase cleavage [123]. Among them, four were deletions and the rest were missense mutations. Two missense mutations, Trp165Cys and Leu173, were reported with two different codon combinations. TM3 was verified to impact the hydrophilic pore formation and to regulate the trimming activity of γ-secretase by interacting with other domains, including TM7. The distance between TM3 and TM7 may be critical in proper γ-secretase function [124]. Several residues were predicted to impact the γ-secretase or other processes (Figure 6), while many were suggested to be in proximity to the APP helical structure and impact the formation of substrate-binding pores, including Trp165, Leu166, Leu173 Trp165, Leu166, and Leu173 [60]. Ser169 may play a role in anchoring the interaction between APP and PSEN1 by forming hydroxyl and carbonyl groups. Leu174 could directly impact the substrate binding [60]. Few residues in TM3 may impact pathogenic mechanisms other than APP cleavage. Mutations in His163 (particularly, His163Arg) may result in the disruption of neurexin processing by γ-secretase and cell adhesion [125]. Furthermore, His163 may also have a high impact on the formation of mitochondria-associated endoplasmic reticulum membrane (MAM). Mutations of His163 may accelerate the process of MAM formation and impair different mechanisms, such as calcium transport, mitochondrial homeostasis, or phospholipid synthesis [126]. In addition to their strong impact on APP cleavage, mutations in Leu166 (particularly, Leu166Pro) could also have a high impact on endosome dysfunctions (enlarged endosomes) by inducing the accumulation of APP β-C-terminal fragments [121]. Leu166 may also be related to calcium balance, since Leu166Pro was suggested to disrupt the ability of PSEN1 to function as a calcium channel and transport calcium to endoplasmic reticulum [127,128,129]. Furthermore, an adjacent mutation, Ser170Phe, seemed also to impair the neuroprotection by disrupting the interactions between trophic factors and GLUN1 receptor [98,130]. Few residues in the region were suggested to impact Notch signaling, including Ser169 [131], Phe176 or Phe177 [60]. Splice site mutations were also found in TM3, since mutations in Gly183 and Glu184 are located in the 3′ end of exon 6 and 5′ end of exon 7, respectively [123,132]. The majority of mutations caused the disease phenotype in patients with the age ranges of the 30s, 40s, or 50s [4]. However, patients with Leu166Pro [127,133], mutations in residue 170 [134,135], and Ser169Leu [131] developed the disease phenotype in their 20s. Three mutations were associated with later disease onset (over 70s), including Ala164Val [136], Ile168Thr [137] or Phe175Ser [138]. EOAD was the main phenotype among patients, but other disease phenotypes also occurred. Patients with Leu166Pro were diagnosed with spastic paraparesis, epilepsy, or ataxia [127,133]. Even though a patient with Leu173Phe (G>C) was diagnosed with EOAD, the initial symptoms were depression and seizures [139]. Gly183Val was associated with frontotemporal dementia without amyloid plaques [132], while one case with Glu184 was also diagnosed with frontal variant AD [140]. FTD-like AD was also observed in patients with Leu174Arg [141]. The family with Glu184Asp had DLB and primary progressive aphasia [142]. Val191Ala in HL3 may not be a pathogenic variant, since it was observed in an individual without any kinds of neurodegenerative diseases [39]. Regarding behavioral changes, psychosis was common among patients, while mutations in residue 163 were related with anxiety [91], delusions [143,144], or psychosis [134]. Additionally, Leu166Val [145], Leu166Arg [77], Phe176Leu [146], Phe177Val [147] were also related with personality dysfunctions. Motor impairments, including myoclonus, tremor or seizures, were also observed among patients with TM3 mutations, for example His163Pro [144], Ser163Pro or Ser170Phe [84].

In the fourth TM domain of PSEN1, 26 mutations have been found. The majority of them were missense mutations, while one indel was reported in this region. The TM4 and TM5 domains face intramembranous hydrophilic milieu and impact the formation of catalytic pores. It was suggested that mutations in the TM4 may impair the Aβ42 generation by altering the intra-and intermolecular interactions. For example, mutations may change the distance between TM4 and TM7 [148]. Furthermore, TM4 may highly impact the intermolecular interactions, since it contains a WNF (Trp203-Asn204-Phe205) sequence, which could impact the presenilin enhancer-2 (PEN2) binding (Figure 7) [16]. Two conservative glycine residues were discovered near the WMF sequence, Gly206 and Gly209 [148,149]. Mutations in Gly206 may directly impact the substrate binding ability of PSEN1 [60]. For example, Gly206Asp was found to lower the PSEN1–PEN2 interaction, resulting in impaired γ-secretase activity and a reduced level of PSEN1 transport from endoplasmic reticulum (ER). Furthermore, mutations in Gly206 may also result in dysfunctions in calcium homeostasis by elevating the calcium levels in ER [150]. Similarly to Gly206, mutations in Gly209 may also directly impact the substrate binding affinity of γ-secretase [60]. Furthermore, Gly209 mutations (especially Gly209Val) could result in abnormal endoproteolytic processing of PSEN1 [94]. Met210 and Ile213 were also found as conserved residues, located at the cytosolic site, impacting the formation of the catalytic core of γ-secretase [149]. Similar to Ile213, His214 was also in proximity to the APP helix and could possibly impact the substrate binding pore formation [60]. Ile213 mutations were associated with reduced neuroprotection by inhibiting several neurotrophic factors, including BDNF [151]. Mutations in Pro218 may impact the PSEN1 splicing [49]. Mutations in TM4 were found in typical EOAD patients, where disease occurred in their 30s, 40s, or 50s [4]. The silent mutation in Pro218 was found in a family over 65 of age, but its pathogenic nature was also questioned [49]. Gly206Ala, found in a large Puerto Rican family, was associated with a wide age range of disease onset. Some patients developed AD as their disease phenotype in their 40s, while some carriers were only diagnosed with AD in their 70s [52]. It may be possible that in the case of Gly206Ala, other genetic factors may impact the disease age onset, including phosphatase synaptojanin-1 (SYNJ1) [152]. Besides EOAD, other disease phenotypes may be associated with mutations in TM4. A patient with Trp203Cys developed bulbar onset ALS without any family history of neurodegeneration [153]. Several patients with Gly206Asp [154] and His214Asn [155] were initially diagnosed with FTD. Additional mutations were associated with behavioral changes, including Gly209Arg [156], Ile213Phe [104], His214Arg [157] and Gly217Arg [158]. EOAD with language dysfunctions were associated with Leu202Phe [105] and Leu219Pro [159]. Gly206Val [160], Gly209Glu [161] and Leu219Arg [162] were associated with both language and behavioral impairment. Motor impairment, including Parkinsonism, gait disturbance, myoclonus, and bradykinesia, also occurred in patients with certain TM4 mutations, including Leu202Phe [163], Ile213Phe [104], His214Asp [36], His214Tyr [155] and Gly217Asp [162]. Among them, Gly217Asp was the only mutation in TM4, which was associated with cotton wool plaques [164]. The α-synuclein pathology was detected in a patient with Ser212Tyr. Seizures were rare among patients with the TM4 mutation, but one member from a French family developed seizures a few years after disease onset [117]. Furthermore, the initial symptom of a patient with Pro218 was visual impairment [165].

In TM5 and HL5, 32 mutations were found; 30 of them were missense variants, while 2 were indels or frameshifts. As mentioned before, along with TM4, TM5 could play a role in the formation of a γ-secretase catalytic site. Furthermore, TM5 could also be involved in PEN2 binding [16]. Two residues in TM5, Met233 and Ile238 were revealed to be located near to Phe177 (in TM3). Residues in TM5 may form a hydrophilic environment around Phe177, and they may play a role in regulating the production of Aβ42 [124]. Mutations in Met233 were related to the accumulation of APP β-C-terminal fragments, which may impair endosomal functions [128]. Furthermore, mutations in Met233 (especially Met233Val) may result in abnormal carboxypeptidase-like cleavage of γ-secretase, but they may not impact the endoproteolytic activity [166]. Phe237 may be involved in the formation of γ-secretase substrate binding pores by generating a hydrophobic pocket [60]. Besides the potential role in amyloid production, Leu235 may also have a critical role in pathogenic mechanisms. Mutations in this residue (particularly, Leu235Val) may impact the neurotransmitter metabolism. Leu235Val was found to accelerate the monoamine-oxidase-A (MAO-A) production, resulting in lower serotonin and noradrenalin expression, leading to depression [167]. Leu235Pro resulted in reduced levels of presynaptic synaptophysin in transgenic mice, suggesting that Leu235 may also impact the appropriate synaptic functions [168]. Mutations in Lys239 resulted in reduced expression of trophic factors involved in cell survival [94]. The frameshift mutation in Pro242 does not impact APP cleavage but may play a role in Notch signaling [169,170]. Furthermore, it may activate the production of several cytokines and chemokines in the case of lipopolysaccharide stimulation. Furthermore, it may activate the production of several cytokines and chemokines in the case of lipopolysaccharide stimulation [170] (Figure 8). TM5 mutations, for Met233Val [171,172], Ser237Cys [173], and Met233Ile [140] were also associated with juvenile disease cases (under 30). The diagnosis was initially FTD in some patients with Leu226Phe [174] or Met233Leu [175]. Behavioral or language impairment occurred in patients with different mutations, such as Gln223Arg [176], Ser230Asn [177], or Leu235Arg [178]. Motor impairments were also quite common among affected patients. The Azeri family with Met233Val initially developed ataxia; later, memory impairments appeared in affected individuals [172]. The affected patients from an American family developed motor skill delay in their childhood, encephalopathy in their 30s, and later they developed AD phenotypes [174]. Parkinsonism was observed in a Spanish family with Leu226Phe [91] and in a Chinese patient with Met233Val [179]. Corticobasal syndrome was also present in a Spanish family with Met233Leu [180], and Lewy bodies appeared in one family with Met233Val from the USA [181]. Interestingly, there was one mutation, Pro232fs (located on HL5), which was not associated with AD or any other neurodegenerative phenotypes. This frameshift mutation resulted in familial acne inversa in Chinese families. The patients displayed hair follicle inflammation and several skin abnormalities, such as abscess in the skin, sinus drains, or scars [169,170].

PSEN1 TM6 is a conserved domain, which contains one of the catalytic aspartate residues (Asp257), that plays a critical role in γ-secretase function and also in endoproteolytic process of PSEN1 [17,182]. Even though no mutations were observed in Asp257, several pathogenic variants (currently 42) were found in the TM6 and its C-terminal loop (HL6). All these mutations were reported as missense mutations. Functional studies predicted that besides Asp257, several residues may also play an important role in PSEN1 function and γ-secretase cleavage. [60]. Ala246 was predicted to be in proximity to the APP helix, and to impact the formation of substrate binding pores. Ala246 may also play a critical role in γ-secretase cleavage-related mechanisms. Mutations in this residue may impair the APP C99 cleavage, resulting in abnormal β-CTF accumulation, and disrupt the endosomes [128,183]. Mutations in Tyr256, Ala260, and Val261 may also impact the C99 cleavage of γ-secretase along with long amyloid production, due to their proximity to Asp257 [184]. Cys263, Pro264, Gly266, Pro267, and Arg278 were suggested to play a role in stabilizing the structural re-arrangement of the PSEN1 protein so as to allow its appropriate binding to the APP helix [184]. Leu271 and Thr274 may play a role in APP binding [60]. Functional studies revealed that mutations in Ala246 (particularly, Ala246Glu) could impair several cellular functions, which are associated with reduced cell differentiation and survival [185]. Mutant Ala246 could disrupt the Notch-Wnt pathways and impair the mitophage, lysosome, and autophage-related pathways [186,187]. Mutations in Pro267 (particularly, Pro267Ser) could also impact pathways, for example by abolishing the cyclophilin B recognition site, resulting in reduced maturation of PSEN1 and abnormal mitochondrial activity [188]. Mutant Leu271 may impact the splicing of PSEN1 exon 8, which could possibly impact the γ-secretase-related mechanisms [189]. Mutations in Ala246 and Glu273 were related to abnormal calcium dynamics and reduced calcium flow to endoplasmic reticulum [129]. Mutations in Pro264 and Glu273 were found to reduce the APOE secretion [61]. Mutant Arg278 (particularly, Arg278Ile) could also result in significant disturbances besides affecting APP cleavage. It may disturb the lipid metabolism by generating disturbances in ApoER2 receptor processing, which could lead to abnormal neuronal migration [190]. This mutation could also result in abnormal Notch cleavage and protein differentiation [94]. Furthermore, the Arg278 mutations may reduce the α-secretase activity by lowering ADAM10 expression (Figure 9) [191]. In terms of disease phenotypes, a few atypical disease cases were associated with TM6 mutations. Mutations were associated with a later onset of disease compared to mutations in the other TM regions. The majority of patients with TM6 mutations developed disease symptoms in their 50s [4]. For example, an American case of Ile249Leu developed an ALS phenotype [153], a patient with Ala275Ser from New Zealand was initially diagnosed with DLB [192], and a family from Germany with Leu272Asp initially developed psychosis [193]. Frontal symptoms (personality changes, language impairment) occurred in patients with Leu262Val [155] and Pro264Leu [194], leading to the initial diagnosis of FTD. Additional AD cases may be associated with behavioral and/or language impairments, including Tyr256Asn [157], Pro264Leu [195] or Arg269Gly [196]. Parkinsonism appeared in patients with Cys263Trp [197] and Leu272Ala [198]. Spasticity and spastic paraparesis may be associated with several mutations, including mutations in Arg278 residue [199,200,201], Gly266Ser [198] or Val261Leu [91]. Myoclonus occurred in patients with several mutations, for example Ile250Val [202], Ile250Ser [203] or Pro264Leu [204]. Cotton wool plaques were observed only in a family with Leu271Val [189].

The large cytosolic loop of PSEN1 can be found between TM6 and TM7. One third of this loop contains a “hydrophobic stretch” (Glu280-Glu299), where several pathogenic mutations were observed. These residues were observed to be highly conserved between the PSEN1 homologues. The region contains sites for PSEN1 endoproteolytic cleavage (near Met292) [205] (Figure 10). Mutations near this region may result in impaired or abolished PSEN1 endoproteolysis. [205]. However, involvement of mutations located in the residues between Gly300 and Gly371 in AD has been questioned. These mutations were found to be weakly conserved among PSEN1 homologues. This region includes a caspase cleavage site at Asp345, which may interact with several PSEN-related proteins, for example β-catenin, members of armadillo protein family, or p0071 proteins. This region may also include PSEN1 phosphorylation sites. However, the large cytosolic loop may not play a significant role in γ-secretase activity, as the deletion of this loop did not impair the production of long amyloid peptides [8,17]. Among the residues in the hydrophobic stretch, Glu280 was suggested to play a key role in APP cleavage. A mutation in this residue may stabilize the interaction between PSEN1 and APP by forming a β-sheet prior to γ-secretase cleavage [60]. Glu280 forms hydrogen bonds with two residues in TM2 (Tyr154 and Tyr159). Mutations in Gly280 may disrupt this contact and reduce the PSEN1 protein production. The altered conformation could result in abnormal APP cleavage and production of long amyloid peptides [206]. Mutations in Glu280 could also impair several cellular processes, resulting in defects in mitochondrial functions, such as calcium homeostasis [207]. It may also be possible that mutant Glu280 may impact Tau phosphorylation [208]. Similar to Arg278, Leu282 may also impact the lipid metabolism, since Leu282Val was found to reduce the surface levels of apoER2 [190]. Leu286 affects substrate binding pore formation of γ-secretase [184]. Furthermore, it may impact calcium dynamics, since Leu286Val was found to disrupt the intracellular calcium metabolism [209]. Similarly to Glu280, Tyr288 may also impact the stabilization of the APP–PSEN1 hybrid. Mutations in residues near the border of exon 8–9 (Ser290, Thr291) of PSEN1 could highly impact PSEN1 splicing and the deletion of exon 9 or exons 9–10 [210]. The abnormal PSEN1 splicing and missing exon 9 could prevent endoproteolytic process of PSEN1, resulting in accumulation of PSEN1 derivatives [211]. The deletion of exon 9 could also impact the APP cleavage and long amyloid peptide generation [212]. Lack of exon 9 could also result in other impairments, such as blocking of ion channels and calcium metabolism [213], lipid metabolism dysfunctions [214] or increasing autophagy (Figure 10) [215]. Patients with mutations located in the N-terminal “hydrophobic stretch” region could develop EOAD, mostly in their 40s or 50s. Mutations in this region, for example Glu280Gly [216], Pro284Leu [217], Leu286Pro [218], c.869-22_869-23ins18 [219], or splice site mutations at Ser290 [219,220,221,222], commonly resulted in EOAD with spastic paraparesis and cotton wool plaques. Mutations which were located in the C-terminal region of the large loop presented a larger scale of phenotypic variability. Trp294Ter mutation occurred in a Chinese family, with patients presenting atypical symptoms, such as retinitis pigmentosa, and acute symptoms resembling viral meningoencephalitis. Visual impairment development began in the teenage years of the affected patients [223]. Most family members with Asp333Gly presented aggressive heart dysfunction, called dilated cardiomyopathy, between the age of 24 and 69. Even though one family member developed AD at the age of 71, it remained unclear whether the Asp333Gly mutation could act as a contributor for the AD phenotype [224]. A patient with the Ser357Ter mutation developed cerebral amyloid angiopathy (CAA), but its disease involvement may not be certain due to the co-existence with a probable pathogenic Arg377Trp mutation [225]. Patients with Pro293Leu [52], Arg352dup [226] and Pro355Ser [227] were diagnosed with FTD or frontal variant AD. Lys311Arg [51] and Glu218Gly may not be fully involved in AD progression, but could act as potential risk factors [228]. The pathogenic nature of Ser365Tyr may also be questioned, since it co-existed with the pathogenic Met146Val mutation [229].

The TM7 domain is also quite sensitive towards mutations, as 27 probable pathogenic mutations appeared in this domain. The majority of them were missense variants, but one frameshift mutation, Gly378fs, was also observed. PSEN1 contains the second catalytic aspartate residue, Asp385. Even though no mutations for Asp385 were found in patients, introduced Asp385 (and Asp257) mutations resulted in elevated levels of CTF-amyloid-β fragments [17,182]. Along with TM6, TM7 could play a critical role in the formation of γ- secretase and in the PSEN1 endoproteolytic process. Besides Asp385, other residues may also play a significant role in γ-secretase activity. Gly378 and Leu381 may impact the APP binding via the formation of three stranded β-sheets prior to APP cleavage [60]. Leu381 may impair the carboxypeptidase-related γ-secretase cleavage, leading to enhanced long amyloid production [167,168,230]. Due to the proximity to Asp385, Gly384 and Phe386 could also be significant in γ-secretase activity. Mutations of these residues may affect the conformation and intramolecular interactions of Asp385 and could result in disturbances inside the active site of γ-secretase. Along with Gly382, Gly384 was found to be part of the conserved GxGD sequence motif, which was suggested to interact with any kind of γ-secretase substrate, including APP or Notch proteins. This domain is sensitive to any kinds of alteration and could result in significant impairment in γ-secretase mechanisms [231,232]. The abnormal enzyme–substrate interactions due to any mutations in the GxGD motif could result in the elevated production of long amyloid peptides 78]. Gly384 was suggested to disturb the carboxypeptidase-like cleavage [168]. Phe388 may impact the APP–PSEN1 interactions by forming a shallow hydrophobic pocket in PSEN1. This pocket may be involved in binding the C-terminal helix of APP [60]. Based on functional studies, some mutations may result in other impairments beyond APP processing. Mutations in Leu381 and Leu392 enhance not only long amyloid production but also PSEN1 endoproteolysis, resulting in lower PSEN1 NTF and Notch processing by γ-secretase [230]. Mutant Gly384 may impair APOE secretion and its localization in cytoplasm [61]. Furthermore, this residue may impact the calcium metabolism, since Gly384Ala may abolish the functions of passive calcium leak channel activity (Figure 11) [61,233]. In terms of phenotypes, the majority of mutations were related to typical EOAD. Young onset disease cases also occurred, with disease symptoms developing under 30 years of age, for example in cases of Leu381Val [230], or Val391Gly [234]. Additional symptoms, such as behavioral disturbances, were relatively common among patients with TM7 mutations, for example in cases with Arg377Trp [235,236], Gly378Glu [237], Tyr389His [143], Ser390Asn [50], Leu392Val [238] or Ala396Thr [155]. Language impairment was less common, occurring in patients with Gly378Arg [238,239] and Pro388Leu [240]. Motor issues, such as Parkinsonism, occurred among patients with Val391Gly [234] and Thr389Ser [84]. Spasticity was reported among patients with Leu381Val [241], or Asn405Ser [242]. One patient with Arg377Trp presented cognitive impairment and cerebral amyloid angiopathy (CAA). This patient had a compound heterozygous mutation of Arg377Trp and Ser357Trp [225]. Patients with Gly378Val had EOAD and corticobasal syndrome [243]. Furthermore, affected members of an American family with Leu381Phe were initially diagnosed with Kufs disease and soon after developed EOAD with ataxia [244].

The TM8 and TM9 domains are located in the C-terminal region of PSEN1. In TM8, 15 mutations were detected. Two (Ala431Val and Ala431Glu) and one (Met457Val) mutations were detected in the HL8 loop and CTF loop, respectively. Crypto-electron microscopy analysis suggested that residues in TM8 (either Cys410 or Cys419) may form disulfide contact with TM1 (Cys92), which may be essential for the γ-secretase function. The TM8 domain contains a conserved AXXXAXXXG domain between Ala409 and Gly417, which could highly impact the catalytic activity of PSEN1. Residues in this motif play a key role in the appropriate conformation of γ-secretase [245,246]. Ala431, located on HL8, could have a putative impact on γ-secretase cleavage. This residue may be in proximity to the β-sheet APP–PSEN1 hybrid and may play a role in APP cleavage [60]. The roles of Leu424 and Ala426 in γ-secretase have not been defined yet, but both residues were verified to be conserved [60]. Furthermore, Leu424 binds several dementia-related mutations, suggesting that it may be a critical residue in γ-secretase processing [247]. Leu418 and Leu420 residues have been suggested to affect APOE secretion, since patients with Leu418Trp and Leu420Arg mutations were associated with lower amyloid production with reduced ApoE levels in the blood compared to non-carriers. These mutations were suggested to abolish the ApoE secretion ability of PSEN1 [61]. Besides the promotion of impaired γ-secretase cleavage, mutant Cys410 (particularly, Cys410Tyr) may also result in abnormal cleavage and transport of Notch-1 [248,249]. Mouse experiments suggested that Cys410 residue may also impact the β-neurexin processing [125]. Mutations in Ile416 and Gly417 residues may also impact the splicing of PSEN1 transcript [250,251]. Ala431 mutations (particularly, Ala431Glu) may play a role in depression by modulating the monoamine oxidase-A (MAO-A) pathway (Figure 12) [167]. Among mutations in TM8, several cases were related with young disease onset—for example, patients with Gly417Ala [251], Leu418Phe [252], Leu424His [171], and Leu424Arg [253] may develop the disease in their 30s. Furthermore, patients with Gly417Ser [254] and Leu418Trp [255] began to show neurodegenerative phenotypes in their 20s and at age 18, respectively. Ile408Thr was related to late disease onset, where the affected family had developed AD in their 70s [256]. Besides typical EOAD, these mutations may be associated with atypical phenotypes; for example, patients with mutations in Gly417 [251,254] and Leu420Arg [257] and Ala31Glu [258] had Parkinsonism. A patient with Val412Ile was diagnosed with FTD [259], while personality changes, language impairment were prominent in patients with Leu418Trp [255] and Leu424Val [260]. An additional atypical case of a Polish patient with the Leu424His mutation, who mainly presented with EOAD but also showcased both DLB and FTD-like symptoms, was observed [174]. One of the cases from the Netherlands with Leu424Arg developed EOAD with CJD and an FTD-like disease course [261]. Interestingly, one patient carrying Ala431Glu had a homozygous form of the mutation. Phenotypes of homo- and heterozygous Ala431Glu seemed to be different. The heterozygous form was related to EOAD with motor and language impairment and seizures, where the affected patients experienced disease in their 40s [258]. The homozygous patient had earlier disease onset (33 years), and experienced learning difficulties and sleep dysfunctions [262]. Cotton wool plaques may not be common among mutations in TM8, since they appeared only in cases with Leu420Arg [257,263].

Ten and one mutations were found in TM9 and CTF fragments, respectively. TMD9 may also impact the γ-secretase cleavage. This domain is highly water-accessible and is located near the catalytic center. Along with TM1, TM6, and TM7, TM9 may play a critical role in forming the hydrophilic catalytic pore. TM9 showed high flexibility and could therefore be crucial in the “gate-open” movement of TM2 and TM6, which could be important to providing accessibility to catalytic aspartates. The TM9 contains a conserved “PALP motif”, which includes the Pro433, Ala434, Leu435, and Pro436 residues. This motif could impact the conformational changes of TM6 and activate the γ-secretase complex [58,59,264]. The PALP domain could act as an APP recognition site by binding residues 718-721 in APP. This binding may stabilize the APP–PSEN1 hybrid prior to proteolytic cleavage [60,265]. Mutations in this area or full deletion of this motif could result in abnormal cleavage of the APP-C83 fragment, leading to increased long amyloid peptide production [60,206]. Mutations in some residues in TM9 including Pro433 and Thr440 may result in abnormal or abolished endoproteolysis and autoproteolysis, respectively [94,266]. Mutations in PALP motif may also reduce or disrupt the Notch-1 proteolysis (Figure 13) [267]. Mutations in TM9 and CTF were associated with early-disease onset cases, especially mutations affecting the PALP motif, such as Pro433Ser [266], Ala434Cys [267] or Pro436Gln [268,269]. The only mutation discovered in CTF (Met457Val) was related to later disease onset, where the proband developed behavioral variant AD at the age of 66 [105]. In terms of disease phenotypes, several atypical cases appeared. One case of Ala434Thr from Korea was diagnosed with Parkinsonism [270], and the Chinese EOAD case with the same mutation was initially diagnosed with schizophrenia [271]. The Spanish case with Pro436Gln had the diagnosis of spasticity and visual agnosia [269]. A family with Thr440del was diagnosed with EOAD, but also presented Parkinsonism and DLB-like phenotypes [272]. Depression was relatively common among mutations, affecting the PALP domain. For example, one patient with Pro433Ser from China [266], a patient with Ala434Thr from Korea [270], a family with Ala434Cys [267] and a patient with Pro436Gln from Australia [267] all presented with depression. Interestingly, one case from the UK of Pro436Gln was associated with mosaicism, since the mutation could not be detected in white blood cells, and instead appeared when DNA from cells of the cerebral cortex was analyzed. This patient developed EOAD with spastic paraparesis at the age of 42 [269]. The pathogenic nature of Ile439Val may be refuted, since it co-existed with the pathogenic Ile413Thr mutation, and did not alter the Aβ42/Aβ40 ratio [94,229].

## 2. Cell Models and Possible Biomarkers of PSEN1 Mutation Pathogenicity

Emerging studies are available on cell models with different PSEN1 mutations [273,274]. Several cell lines can be used, including human embryonic kidney cells 293 (HEK293), green monkey kidney cells (COS-1), Chinese hamster ovary cells (CHO), neuroblastoma (N2a), SH-SY5Y, human neuroglioma (H4), fibroblasts, and human induced pluripotent stem cells (iPSC) [188,274,275,276,277,278]. One of the earliest functional studies on PSEN1 mutations was performed by Murayama et al. (1999) [276]. They introduced 28 familial EOAD-related PSEN1 mutations into COS-1 cells using site-directed mutagenesis and measured the Aβ42 levels and Aβ42/40 ratios. Even though the majority of mutations were associated with elevated amyloid levels, Aβ42 levels did not correlate with the age of disease onset. This poor correlation may be explained by the significant biological differences between COS-1 cells and the human brain. Additional genetic and environmental factors could also explain the poor correlation of the above biomarkers in CHO cells with expressing normal or mutant PSEN1 (Asn135Asp, His163Arg, Met239Thr, and Gly384Ala) [276,277]. Mutations were generated using plasmids containing normal PSEN1 (and PSEN2), and cDNA was used as a template for certain mutations. They measured Aβ42/Aβ40 levels and analyzed cell-free and intracellular amyloid production. This study revealed the differences between cell-free and intracellular PSEN1 mutations. Furthermore, some PSEN1 mutations reduced Aβ40 production, while others did not, and all examined PSEN1 mutations were related to higher Aβ42 levels. These findings suggest that the mutations may act through distinct mechanisms [277]. Shioi et al. introduced PSEN1 mutations into CHO cell lines and found that several pathogenic PSEN1 mutations (such as Val82Leu, Leu250Ser, or Cys263Arg) were not associated with elevated Aβ42 levels or Aβ42/40 ratio. This study also suggested that some of the PSEN1 mutations could impact neurodegeneration through other mechanisms, which may be independent of amyloid processing, such as Tau phosphorylation or Akt signaling [278]. Houlden et al. (2000) cloned different mutations (exon 9 deletion, Pro436Gln) that are associated with EOAD and spastic paraparesis into H4 glioma cells. These findings reveal that mutations, which also present spastic paraparesis as a disease phenotype, may have large effects on amyloid production. However, this study did not rule out the potential roles of other genetic and environmental risk modifiers [61]. Kumar-Singh et al. (2006) co-introduced eight PSEN1 mutations with Swedish APP mutations (Lys670Asn and Met671Leu) into HEK293 cell lines using plasmid methods. This study revealed a strong correlation between age at disease onset and an elevated Aβ42/40 ratio, higher Aβ42, and lower Aβ40 levels [48]. Sun et al. (2017) performed an extensive functional study on 138 PSEN1 mutations and used the CRISPR-Cas9 transfection method to generate N2a mutant cell lines. This study did not find a correlation between the Aβ42/40 ratio and the age at disease onset. The majority of mutant cells showed reduced levels of Aβ42 and Aβ40, and approximately 10% of the mutations were related to the Aβ42/40 ratio. This study suggests that the amyloid hypothesis may not be the only hypothesis in the case of AD onset, and other risk factors should be examined [94]. Hsu et al. used N2a cell lines to verify several VUS in EOAD-causing genes, including PSEN1. This study identified 19 mutations (11 PSEN1 mutations) as probable pathogenic variants [39]. Li et al. (2016) analyzed several PSEN1 mutations in HEK293 cell lines, revealing that in addition to the Aβ42/40 ratio, the cleavage of Aβ43 to Aβ40 and Aβ42 to Aβ40 may be reduced in the case of PSEN1 mutations. Some PSEN1 mutations could affect the Aβ42/40 ratio through the Aβ49/Aβ40 and Aβ48/38 ratios, whereas some mutations could only affect one of them [166].

The majority of PSEN1 mutations result in an elevated Aβ42/Aβ40 ratio by elevating Aβ42 production or reducing Aβ40 production. Increased levels of Aβ42 and/or reduced levels of short Aβ40 in cell and animal models may provide strong proof of the pathogenic nature of the mutation [273]. In addition to Aβ42 and Aβ40, other products may reflect the altered γ-secretase process. APP processing may be initiated by β-secretase cleavage, which results in the release of the N-terminal region of the protein. The membrane-bound remnant of APP, called C99, is processed by γ-secretase. This process could result in long amyloid peptides with endopeptidase activity (43–51 bp), which could be cleaved to Aβ42 and Aβ40 through carboxypeptidase activity [279]. Few mutations may reduce the endo- or carboxypeptidase activity of γ-secretase, resulting in the generation of long amyloid peptides, including Aβ43, Aβ46, or Aβ48 [280]. Aβ43 has also been suggested to act as a toxic peptide, generated regularly by PSEN1 mutations, such as Val261Phe or Arg278Ile. Production of Aβ43 may be a useful marker, indicating the relation between the mutation and the impaired endopeptidase activity [184]. Besides Aβ40, other short amyloids, such as Aβ38 and Aβ37, may also be reduced in the case of PSEN1 mutations (e.g., PSEN1 Ala97Val or Val89Leu). Reduced levels of short amyloids may also be a marker of lower amyloid trimming activity and reduced γ-secretase processivity. Furthermore, the reduced ratio of combined short amyloids vs. combined long amyloids ((Aβ38 + Aβ37 + Aβ40)/(Aβ42 + Aβ43)) may be a useful marker of disease pathogenicity [73,281].

## 3. Discussion

PSEN1 has been identified as the most common causative gene for early onset AD. To date, more than 300 mutations have been reported in PSEN1 that potentially play a role in neurodegenerative pathways. Most PSEN1 mutations are heterozygous and follow an autosomal dominant inheritance pattern. However, homozygous forms of PSEN1 mutations, such as Ala431Glu or Glu280Ala, have also been observed. The homozygous forms of PSEN1 mutations may be associated with a more aggressive disease phenotype than the heterozygous mutations [262]. A compound heterozygous case of PSEN1 was also discovered, wherein the patient carried a STOP codon mutation at residue Ser357 and the Arg377Trp mutation. However, it remains unclear whether these two mutations affect disease onset [225].

PSEN1 mutations could result in neurodegeneration through both gain-of-function [282] or loss-of-function mechanisms [283]. Hardy and Higgins (1992) proposed the amyloid hypothesis, suggesting that the accumulated amyloid peptides may be the main causative factors for AD related neurodegeneration. Additionally, mutant PSEN1 mutations (such as Met84Val, Leu85Pro, His163Arg, His163Pro, Met233Leu) could enhance the APP processing and amyloid peptide (Aβ42) generation. However, this study also raised concerns about the amyloid hypothesis. For example, amyloid plaques in the brain may not correlate with the degree of neurodegeneration. Additionally, amyloid production in cell cultures may not correlate with the age of onset or disease phenotypes [40,282,284,285,286]. Later, the amyloid hypothesis was refuted further. It was revealed that elevated Aβ42 may not be the only factor leading to PSEN1-related neurodegeneration. For example, PSEN1 mutations may result in earlier disease onset, compared to mutations in APP, even though Aβ42/40 ratio may not be as significant as expected [286]. Additionally, amyloid overproduction did not result in significant neurodegeneration in mouse models [285,286,287]. These findings suggest that the loss of PSEN1 functions may play a crucial role in AD progression. PSEN knockout mice presented a significant degree of neurodegeneration; however, the level of both Aβ42 and Aβ40 was reduced. PSEN knockout could result in an elevated degree of neuroinflammation, reduced neuroprotection and an increased degree of apoptosis [288,289,290]. Shen and Kelleher (2007) proposed the presenilin hypothesis of AD, which may provide an alternative view of disease pathogenesis. The loss of essential PSEN1 (and PSEN2) functions may result in AD-related neurodegeneration. Several PSEN1 mutations (Gly209Arg, Gly209Val, Leu235Pro, Cys410Tyr, Leu435Phe) may not significantly impact the Aβ42 levels (or even reduce it), but they may reduce (or abrogate) the Aβ40 production significantly. Pathogenic PSEN1 mutations may impair the γ-secretase functions through dominant-negative mechanisms. Elevated amyloid levels may also inhibit further the γ-secretase functions. Loss of PSEN activity may result in abnormalities in synaptic functions, leading to neuronal loss, Tau hyperphosphorylation and dementia. Further studies are also needed on the PSEN hypothesis. For example, APP mutations may not inhibit γ-secretase related pathways. Additionally, not all PSEN1 mutations could result in a Tau-related pathology [40,283,291]. Besides amyloid and PSEN hypothesis, other AD-hypotheses were also described, such as the Tau hypothesis, the inflammation hypothesis, and the cholinergic and oxidative stress hypothesis, confirming that AD is a very complex disease. There is no absolute hypothesis available on AD progression, but all possible hypotheses could provide an explanation for AD progression and help drug development [282,291,292].

As mentioned before, PSEN1 mutations may impair γ-secretase mechanisms, resulting in impaired amyloid production and elevated long amyloid/short amyloid (typically Aβ42/40) ratio [6]. Furthermore, PSEN1 could affect other mechanisms, such as calcium homeostasis or Notch signaling [6]. Compared to APP- and PSEN2-related cases, EOAD patients with PSEN1 mutations may have an earlier disease onset [293]. Additionally, in the case of APP mutations, memory decline is the initial symptom, whereas PSEN1 mutations may lead to atypical phenotypes, such as seizures or motor-behavioral or language dysfunctions. The mutation location may impact the PSEN1 phenotype. With regard to the atypical symptoms in cases of PSEN1 mutations, the location of the mutation may be important. Mutations located in the N-terminal region (before codon 200) may more frequently represent myoclonus, seizures, visuospatial symptoms, or spasticity. Meanwhile, among patients who carry mutations in the C-terminal area (after codon 200), cotton wool plaques and amyloid antipathy may be relatively common [292]. Brain imaging has revealed that APP mutations may result in a higher degree of hippocampal atrophy. PSEN1 mutations may also affect other brain areas such as the neocortex or white matter [77]. The phenotypic diversity of PSEN1 mutations may also be related to its involvement in γ-secretase function. PSEN1 impacts the formation of the γ-secretase catalytic subunit, and in addition to APP processing, it could impact multiple substrate cleavage (β-catenin, Notch). PSEN1 mutations can affect endopeptidase and carboxylpeptidase activities, which may result in different types of neurodegenerative process [284,294]. Interestingly, patients with the same PSEN1 mutations may have different ages of onset or clinical course. For example, EOAD with PSEN1 Thr119Ile has been discovered in multiple patients with a wide age range of ages of disease onset (between 49 and 71 years); some cases developed memory decline only, while in others, personality changes and psychological symptoms were present. It is possible that other genetic factors may impact disease onset and clinical course, for example, variants in SORL1 or ABCA7 [82,295]. Risk modifiers were also analyzed in a large Caribbean family with the Gly206Ala mutation. Some variants may result in a delayed age of onset, for example, variants in SORBS2, SH3RF3 or NPHP1 [296]. The Paisa mutation (PSEN1 Glu280Ala) is another common EOAD-causing factor, which may possibly be affected by genetic modifiers. APOE Arg145Ser (or the Christchurch mutation) was verified to be a neuroprotective factor against EOAD with PSEN1 Glu280Ala [296]. Additional genetic variants may affect the age of disease onset in the case of the Glu280Ala mutation. For example, variants in SLC9C1, CSN1S1, and LOXL may be associated with a later disease onset, whereas a homozygous DHRS4L2 variant (rs2273946) may result in earlier disease onset [297]. These findings reveal that risk modifiers could impact the age of onset of EOAD related to PSEN1 mutations and may also impact the clinical course of the disease [298].

Next-generation sequencing technologies reduce the cost and time of gene analysis and help to accelerate the discovery of disease-related mutations in causative or risk genes including PSEN1 [35]. However, it may be challenging to determine whether PSEN1 mutations could affect disease onset. Based on the ACMG-AMP guidelines, several PSEN1 mutations could not be classified. Several limitations exist, which may result in challenges in mutation classification. For example, several mutations (such as His163Pro, Leu232Pro, Gly417Ala) were present in only one patient. Furthermore, family members often refuse genetic testing and segregation can, therefore, not be proven. Additionally, mutations may not be confirmed by in vitro or functional studies [38,39]. Functional analysis is a promising approach in genetics and drug discovery studies. However, in vitro cell and in vivo animal studies require special equipment, an appropriate environment, and ethical approval. Other issues with genetic studies may include potential technical errors or off-target mutagenesis [273]. However, recent emerging studies on genome editing could reveal the pathological mechanisms of diseases, including AD [299].

Another issue is that even though several cases of EOAD are related to PSEN1, the majority of AD cases may not be explained by classical EOAD genes. Emerging studies have shed light on somatic mutations in brain diseases and brain aging, with one study particularly focusing on somatic PSEN1 mutations. In one patient with PSEN1 Pro436Gln, the mutation could not be detected in white blood cells but was found in brain cells [269]. However, to date, only one study has been conducted on somatic PSEN1 mutations. It has been suggested that somatic APP, PSEN1, and PSEN2 variants are rare in sporadic EOAD [300]. An additional limitation of studying somatic mutations is the availability of patient brain tissue [301].

In conclusion, the significance of PSEN1 in EOAD has been verified. As a part of γ-secretase, the PSEN1 protein can affect the processing of multiple substrates, including APP, Notch, and β-catenin. PSEN1 mutations could be related to different phenotypes (seizure, behavioral, language, or motor impairment), which may be related to the impairment of different mechanisms and the induction of neurodegeneration in different brain areas. Although rare, cases with PSEN1 mutations should be taken into consideration seriously. Emerging in vitro and in vivo studies are available to verify the role of PSEN1 mutations in AD, which could improve disease diagnosis and therapeutic strategies.

## Figures and Tables

**Figure 1 ijms-23-10970-f001:**
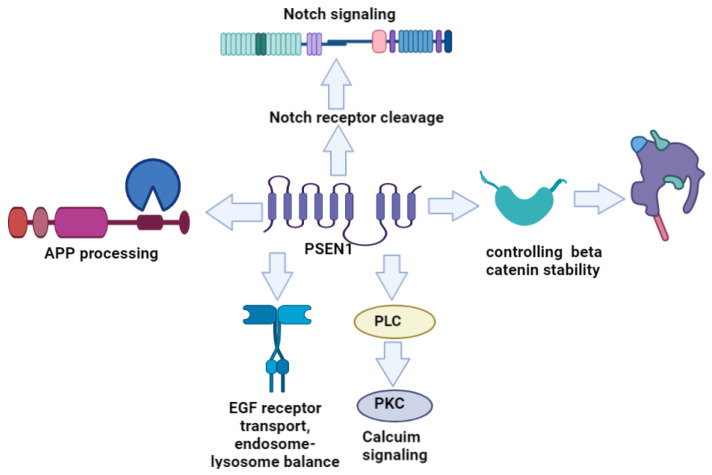
Pathways, in which PSEN1 may be involved.

**Figure 2 ijms-23-10970-f002:**
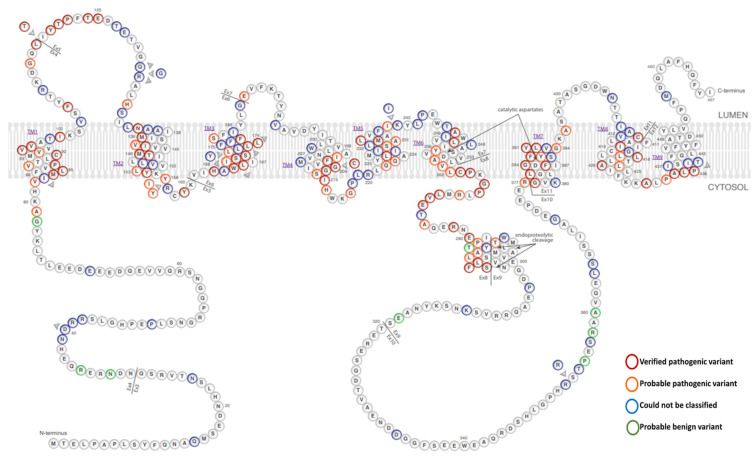
Mutation sites in PSEN1 protein.

**Figure 3 ijms-23-10970-f003:**
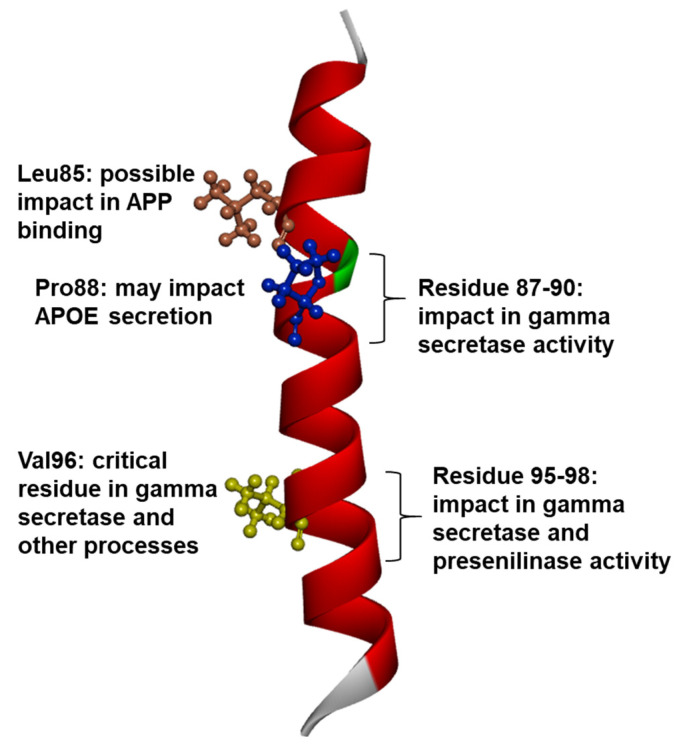
Significant residues in PSN1 TM1.

**Figure 4 ijms-23-10970-f004:**
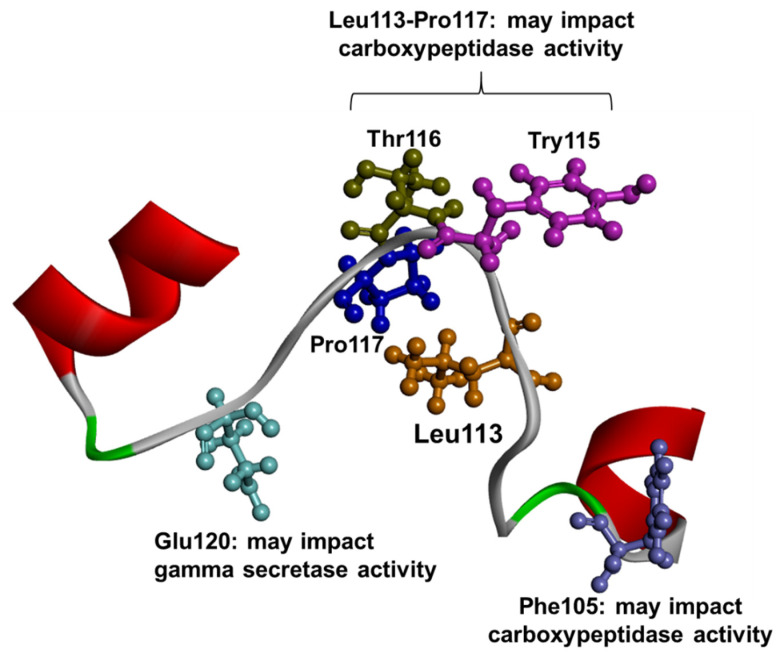
Residues in HL1 which may be critical in γ-secretase production.

**Figure 5 ijms-23-10970-f005:**
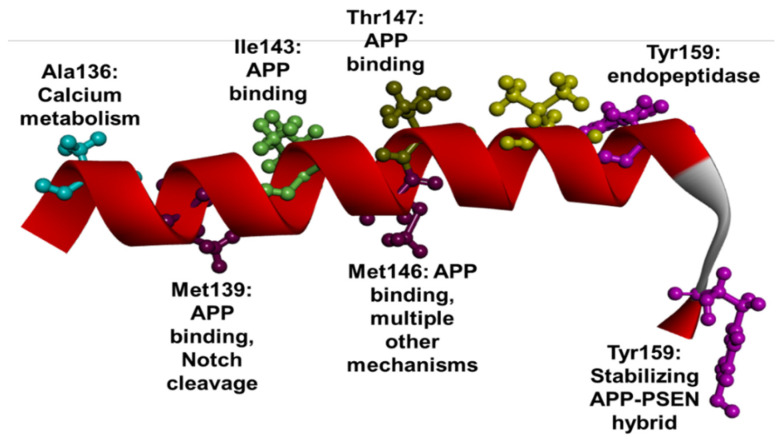
Potential significant residues in TM2 and HL2.

**Figure 6 ijms-23-10970-f006:**
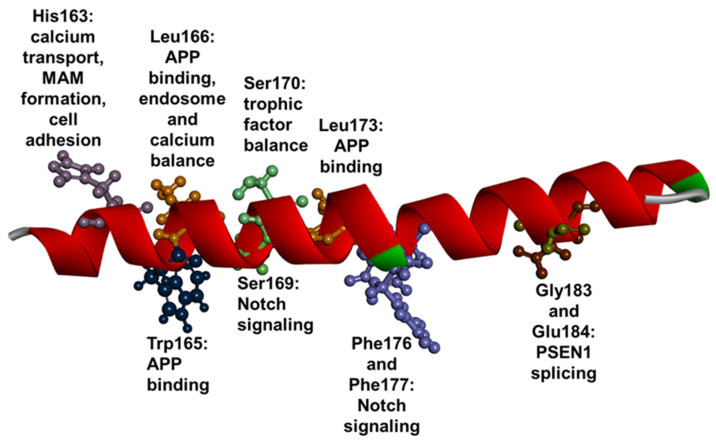
Significant residues in TM3.

**Figure 7 ijms-23-10970-f007:**
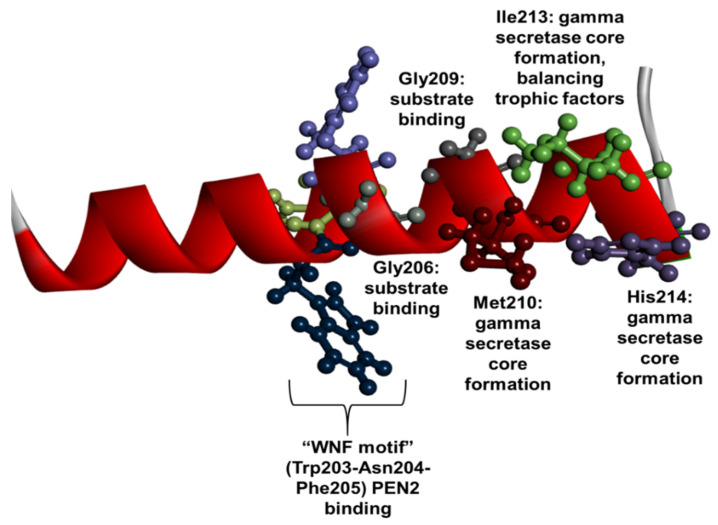
Possible significant residues in TM4.

**Figure 8 ijms-23-10970-f008:**
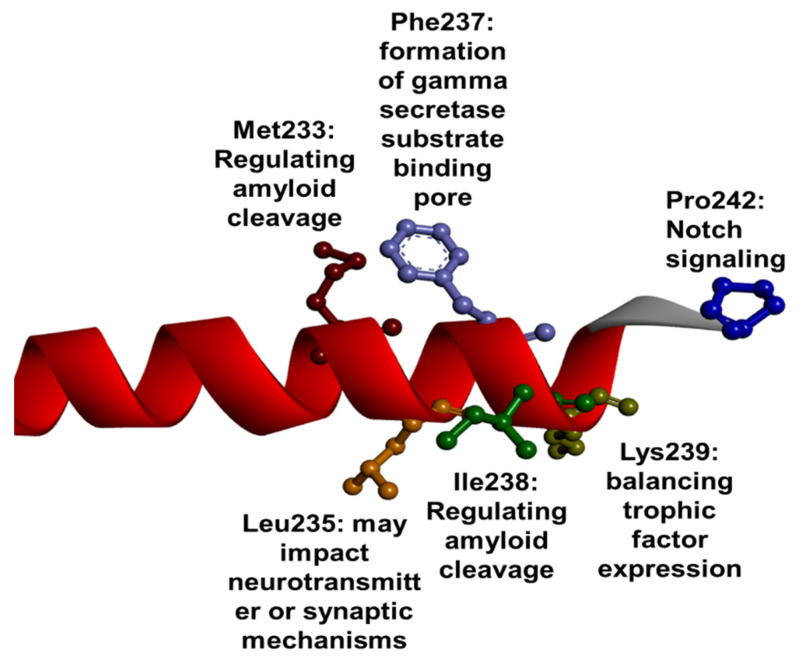
Significant residues in TM5 and HL5.

**Figure 9 ijms-23-10970-f009:**
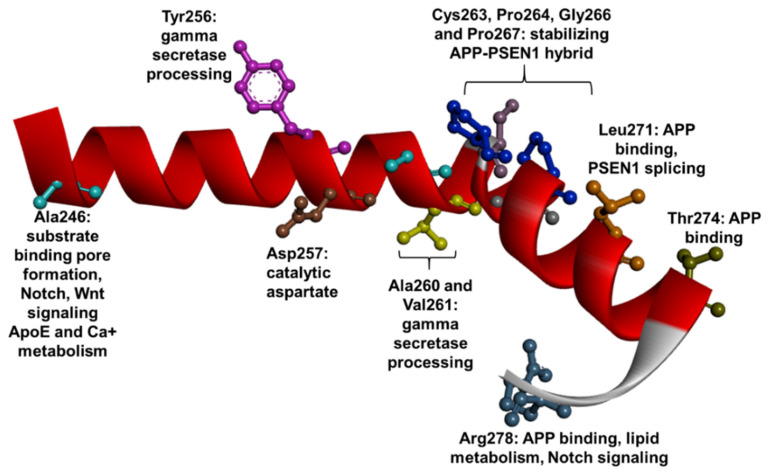
Significant residues in TM6 and HL6.

**Figure 10 ijms-23-10970-f010:**
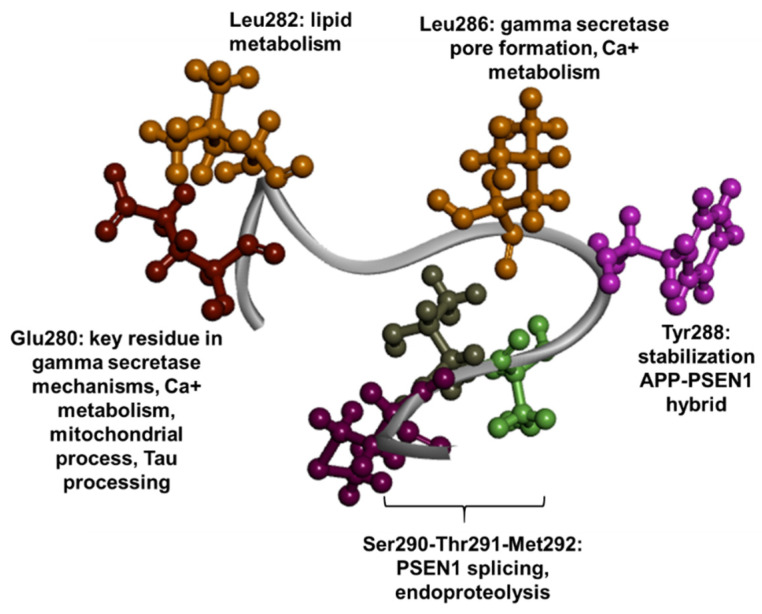
Significant residues located in the hydrophobic stretch of large loop.

**Figure 11 ijms-23-10970-f011:**
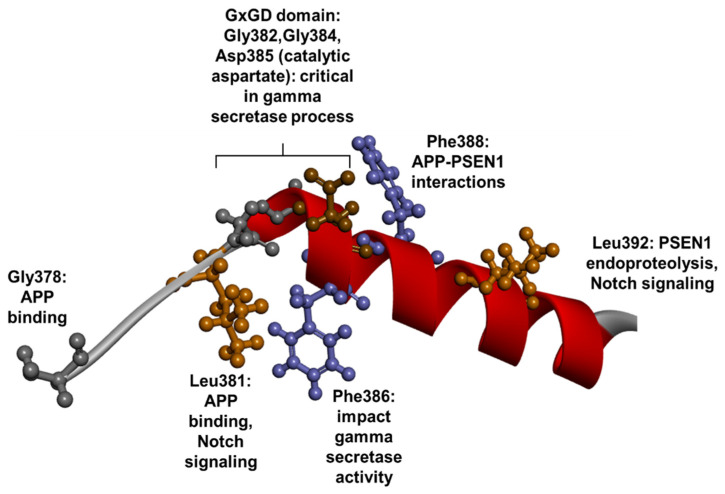
Significant residues in TM7 and HL7.

**Figure 12 ijms-23-10970-f012:**
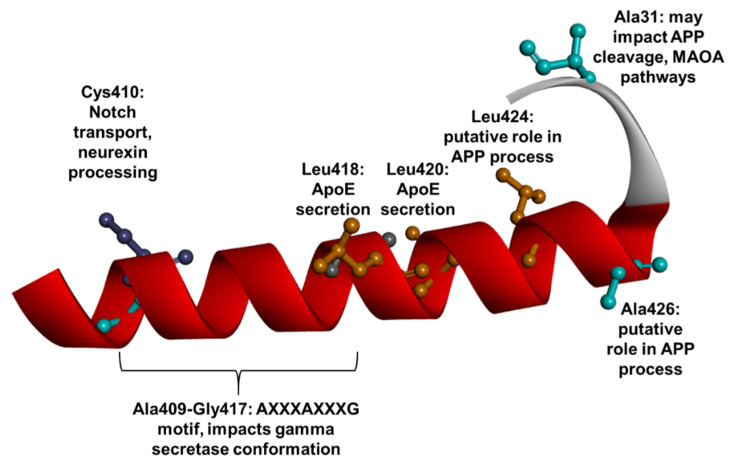
Significant residues located in TM8 and HL8.

**Figure 13 ijms-23-10970-f013:**
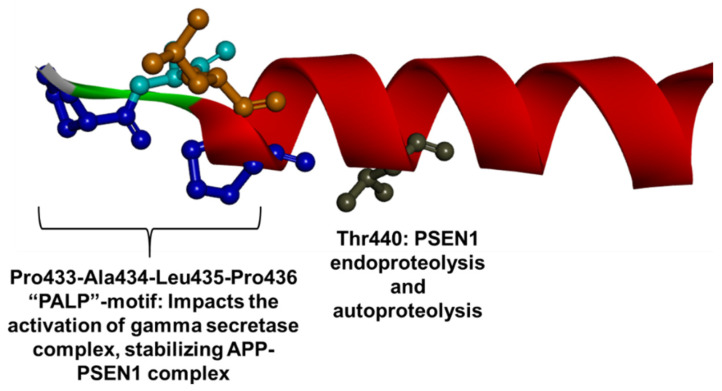
Significant residues located in TM9.

**Table 1 ijms-23-10970-t001:** Categorizing EOAD-related (including PSEN1) mutations based on the ACMG-AMP guidelines [37,38,39].

Criteria	Weighting	Proof
Pathogenic	Strong	Mutation was found in several unrelated disease cases Functional (in vivo or in vitro) studies are well established, and revealed damaging mechanisms Prevalence is higher in patients than controls De novo case of mutation, both parents confirmed as non-carriers Mutation segregates with disease, which is confirmed
	Moderate	Mutation is located in a “hot spot” residue, or functional domain/affects a critical residue Missing or very rare in reference database (GnomAD, 1000Genomes) Novel variant in a residue where pathogenic variant was reported before De novo variants, but parents were not confirmed as non-carriers Mutation segregates with disease, but cannot be confirmed
	Supporting	Mutation found in a gene, where missense variants are common in diseases In silico predictions confirm it as damaging—for example, CADD scores more than 20. Phenotype of patient is specific for a disease, no other pathogenic mutations appeared in proband. Mutation may affect a conserved residue (among vertebrates) Frameshift variant/indel in non-repeat region, stopgain or stoploss variant
Benign	Supporting	Mutation frequency is more than 5% in reference databases Structure predictions revealed the mutations to be non-damaging (CADD scores lower than 20). Residue may not be conserved
	Strong	Other disease-associated mutations were found in the patients Mutation frequency in controls may be higher than in patients Mutation does not segregate with disease Observed in at least one healthy age matched individual Multiple functional studies did not reveal increased Aβ42/40 ratio or any pathogenic mechanisms
	Protective (?)	Functional studies revealed reduced Aβ42/40 levels, compared to controls

## Data Availability

Not applicable.

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
