# Peer review of "Genetics, Functions, and Clinical Impact of Presenilin-1 (PSEN1) Gene"

_ijms, 2022, doi:10.3390/ijms231810970_

Round 1

Reviewer 1 Report

The manuscript by Bagaria et al. is both content-extensive and well-written. Also, it is worth noticing the high quality of the figures, which the authors clearly took great efforts into making. Generally, there is only a few typo errors and wording inconsistencies need to be corrected before allowing publication (will be mentioned later).

P(age) 1, abstract section: I would recommend the consistence use of greek symbols through the articles. e.g.: gamma secretase was used but also b-catenin. Editors may be able to provide some guidance for the authors on which format to use;

Page 6, section 1.3: Hek293 -> HEK293;

Page 7, section 1.3: both TM-1 and TM1 were used to describe the transmembrane domain 1 region, if I understand correctly. Better use one unified nomenclature to prevent potential confusions;

Page 7, section 1.3: ...develop disease onset in their 30s. Missing a full stop;

Page 7, section 1.3: ...spratic paraparesis alongside visual: need a space between "alongside" and "visual";

Page 15, section 1.3: ...and impair the mitophagy-, lysosome-, and authphagy-related pathways. Change to "mitophage, lysosome, and authphage-related pathways";

Page 17, section 1.3, ...levels of CTF-amyloid beta fragments: beta or b?

Author Response

Comments and Suggestions for Authors
The manuscript by Bagaria et al. is both content-extensive and well-written. Also, it is worth noticing the high quality of the figures, which the authors clearly took great efforts into making. Generally, there is only a few typo errors and wording inconsistencies need to be corrected before allowing publication (will be mentioned later).

Thank you very much for the positive feedback. The typos have been fixed.

P(age) 1, abstract section: I would recommend the consistence use of greek symbols through the articles. e.g.: gamma secretase was used but also b-catenin. Editors may be able to provide some guidance for the authors on which format to use;

Thank you, they were unified into Greek alphabet.

Page 6, section 1.3: Hek293 -> HEK293;

Issue was fixed

Page 7, section 1.3: both TM-1 and TM1 were used to describe the transmembrane domain 1 region, if I understand correctly. Better use one unified nomenclature to prevent potential confusions;

Thank you, they were unified

Page 7, section 1.3: ...develop disease onset in their 30s. Missing a full stop;

Thank you, issue has been fixed “carriers may develop disease phenotypes in their 30s.”

Page 7, section 1.3: ...spratic paraparesis alongside visual: need a space between "alongside" and "visual";
Thank you, issue was fixed

Page 15, section 1.3: ...and impair the mitophagy-, lysosome-, and authphagy-related pathways. Change to "mitophage, lysosome, and authphage-related pathways";

Thank you, issue has been fixed

Page 17, section 1.3, ...levels of CTF-amyloid beta fragments: beta or b?

The issue has been fixed: elevated levels of CTF-amyloid-b fragments

Reviewer 2 Report

This manuscript is a review article introducing the role of PSEN1 in gamma secretase and the relationship between the clinical phenotypes of EOAD and the mutations of PSEN1. The article is well-written and well documented. It might be accepted for publication in the present form.

Author Response

This manuscript is a review article introducing the role of PSEN1 in gamma secretase and the relationship between the clinical phenotypes of EOAD and the mutations of PSEN1. The article is well-written and well documented. It might be accepted for publication in the present form.

Thank you very much for the positive feedback. We really appreciate it.

Reviewer 3 Report

Re: ijms-1845693

Numerous mutations causing early-onset familial Alzheimer’s disease are identified in the genes encoding APP and Presenilin (PSEN). PSEN is the catalytic component of γ-secretase responsible for the cleavage of APP, suggesting that APP processing by γ-secretase plays a pivotal role in the pathogenesis of early-onset AD. Furthermore, the mutations in the genes encoding PSEN are also linked to epileptic seizures and frontotemporal dementia. Therefore, it is obvious that a review article in which the genetics, functions, and clinical impact of PSEN are comprehensively discussed is highly demanded.

The reviewer could positively find how much effort the authors (presumably the first author) dedicated to preparing this review article. In particular, section 1.2. --PSEN1 mutations and their classification-- is very informative, and section 1.3. --PSEN1 mutations and significant residues at each domain-- is impressive as I have never seen such a review comprehensively summarizing the clinical impact(s) and potential pathogenic molecular mechanism(s) of the over 300 mutations.

However, the reviewer found that this article includes many erroneous phrases. In addition, the reference list does not adequately cover relevant original works; rather often cites previously published review articles that do not specifically discuss the relevant studies. Many sentences describing previous findings do not adequately refer to citation(s). Furthermore, there are many errors in formatting (e.g., space between letters).

Here are a few examples, but more issues would likely be found:

1.     Introduction: The authors state “AD has different neuropathological hallmarks, including extracellular amyloid plaques, cerebral amyloid angiopathy (CAA), intracellular neurofibrillary tangles (NFTs), and the loss of neurons and synapses [3].” Although I agree that CAA frequently co-occurs with AD, whether we should count it as one of the pathological hallmarks of AD is unclear.

2.     Introduction: The authors state “Early onset AD (EOAD) represents 5-10% of all AD cases and can occur under 65 years of age.” 10% is too much; many papers report it within the 1-5% range.

3.     Section 1.1. PSEN1 structure and functions: The authors state “The N-terminal loop and the large hydrophilic loop are located in the cytosol region, while the C-terminal loop is located in the extracellular space. The large hydrophilic loop also contains a membrane-associated area.” The N-terminal (1-82 a.a.) and C-terminal regions (455-467 a.a.) do not form loop structures.

4.     Section 1.1. PSEN1 structure and functions: The authors state that “As part of the gamma secretase complex, PSEN1 and PSEN2 can interact with various proteins, including nicastrin (Nct), PSEN enhancer 2 (PEN2), and anterior pharynx 1 (APH-1).” I was somewhat uncomfortable with this sentence; a proper way could be, “As part of the γ-secretase complex, PSEN1 or PSEN2 forms a complex with nicastrin (Nct), PSEN enhancer 2 (PEN2), and anterior pharynx 1 (APH-1). Furthermore, the γ-secretase complex is known to interact with various other proteins, including…” (for instance, see binding proteins uncovered in Wakabayashi et al. 2009).

Wakabayashi T, Craessaerts K, Bammens L, Bentahir M, Borgions F, Herdewijn P, Staes A, Timmerman E, Vandekerckhove J, Rubinstein E, Boucheix C, Gevaert K, De Strooper B. Analysis of the gamma-secretase interactome and validation of its association with tetraspanin-enriched microdomains. Nat Cell Biol. 2009 Nov;11(11):1340-6.

5.     Section 1.1. PSEN1 structure and functions: The authors state “Knockout of the PSEN1 gene in mice resulted in elevated production of amyloid beta peptide (Ab42) due to abnormal processing of APP protein by gamma secretase, but it did not affect the cleavage of alpha and beta secretase.” This sentence is incorrect. De Strooper et al. (1998) showed in Fig. 3 that PSEN1 knockout results in a significant decrease in Abeta42 levels both in the medium and neuronal culture lysate. Furthermore, the authors state “A study by Yonemura et al. examined the gamma secretase activity of PSEN1 and PSEN2 in yeast models. PSEN1- gamma secretase represented significantly higher activity, compared to PSEN2 related gamma secretase in beta-galactosidase assay.” However, this study compared the activity of PSEN1 or PSEN2 per one γ-secretase complex and found that PSEN1/γ-secretase and PSEN2/γ-secretase exhibit similar activities. These apparent misunderstandings of previous works of literature raise a serious concern about whether this review article adequately delivers previous scientific findings to the readers.

6.     Section 1.1. PSEN1 structure and functions: The authors state “The TM6 and TM7 domains contain two catalytic aspartates.”, which is correct; however, the citation [16] is incorrect. Wolfe et al., 1999, for the first time, reported that the two aspartates, a.a. number 257 and 385, are required for presenilin endoproteolysis and γ-secretase activity.

Wolfe MS, Xia W, Ostaszewski BL, Diehl TS, Kimberly WT, Selkoe DJ. Two transmembrane aspartates in presenilin-1 required for presenilin endoproteolysis and gamma-secretase activity. Nature. 1999 Apr 8;398(6727):513-7.

Similarly, the reviewer found that there are many references not adequately cited. For example, [18] is incorrect, and I believe it should be De Strooper et al., 1999. This is an example, and the authors should fully ensure that the references they cited are correct.

De Strooper B, Annaert W, Cupers P, Saftig P, Craessaerts K, Mumm JS, Schroeter EH, Schrijvers V, Wolfe MS, Ray WJ, Goate A, Kopan R. A presenilin-1-dependent gamma-secretase-like protease mediates release of Notch intracellular domain. Nature. 1999 Apr 8;398(6727):518-22.

7.     Section 1.1. PSEN1 structure and functions: The authors state “Notch signaling has been verified to determine cellular fate and control cell differentiation and division.” This is correct; however, the authors should refer to citations. The reviewer also found many other sentences missing citation(s). For instance, the sentence “PSEN1 may also be involved in Wnt signaling by controlling β-catenin stability.” needs to cite Zhang et al., 1998. Again, this is only a few examples and I expect more; the authors need re-check of the entire manuscript to ensure if necessary references are adequately cited.

Zhang Z, Hartmann H, Do VM, Abramowski D, Sturchler-Pierrat C, Staufenbiel M, Sommer B, van de Wetering M, Clevers H, Saftig P, De Strooper B, He X, Yankner BA. Destabilization of beta-catenin by mutations in presenilin-1 potentiates neuronal apoptosis. Nature. 1998 Oct 15;395(6703):698-702.

Taken together, the reviewer highly recommends that the authors extensively revise the article by re-ensuring that 1) citation(s) is included when the authors describe scientific consensus, 2) the authors cite the original research articles, and importantly 3) the authors adequately summarize what previous original articles reported.

Author Response

Numerous mutations causing early-onset familial Alzheimer’s disease are identified in the genes encoding APP and Presenilin (PSEN). PSEN is the catalytic component of γ-secretase responsible for the cleavage of APP, suggesting that APP processing by γ-secretase plays a pivotal role in the pathogenesis of early-onset AD. Furthermore, the mutations in the genes encoding PSEN are also linked to epileptic seizures and frontotemporal dementia. Therefore, it is obvious that a review article in which the genetics, functions, and clinical impact of PSEN are comprehensively discussed is highly demanded.

The reviewer could positively find how much effort the authors (presumably the first author) dedicated to preparing this review article. In particular, section 1.2. --PSEN1 mutations and their classification-- is very informative, and section 1.3. --PSEN1 mutations and significant residues at each domain-- is impressive as I have never seen such a review comprehensively summarizing the clinical impact(s) and potential pathogenic molecular mechanism(s) of the over 300 mutations.

However, the reviewer found that this article includes many erroneous phrases. In addition, the reference list does not adequately cover relevant original works; rather often cites previously published review articles that do not specifically discuss the relevant studies. Many sentences describing previous findings do not adequately refer to citation(s). Furthermore, there are many errors in formatting (e.g., space between letters).
Thank you very much for the positive feedback and constructive critics. We revised our manuscript according to your suggestions.

Here are a few examples, but more issues would likely be found:

1.     Introduction: The authors state “AD has different neuropathological hallmarks, including extracellular amyloid plaques, cerebral amyloid angiopathy (CAA), intracellular neurofibrillary tangles (NFTs), and the loss of neurons and synapses [3].” Although I agree that CAA frequently co-occurs with AD, whether we should count it as one of the pathological hallmarks of AD is unclear.

Thank you very much, we changed this sentence

“AD has different neuropathological hallmarks, including extracellular amyloid plaques, intracellular neurofibrillary tangles (NFTs), and the loss of neurons and synapses. Additional atypical neuropathological phenotypes, such as cerebral amyloid angiopathy (CAA), cot-ton-wool plaques, Lewy bodies or Pick’s bodies may also co-occur in patients with AD”

2.     Introduction: The authors state “Early onset AD (EOAD) represents 5-10% of all AD cases and can occur under 65 years of age.” 10% is too much; many papers report it within the 1-5% range.
Thank you, issue has been fixed.

  1.     Section 1.1. PSEN1 structure and functions: The authors state “The N-terminal loop and the large hydrophilic loop are located in the cytosol region, while the C-terminal loop is located in the extracellular space. The large hydrophilic loop also contains a membrane-associated area.” The N-terminal (1-82 a.a.) and C-terminal regions (455-467 a.a.) do not form loop structures.

Thank you, issue has been fixed: “The N-terminal fragment and the large hydrophilic loop are located in the cytosol region, while the C-terminal fragment is located in the extracellular space. The large hydrophilic loop also contains a membrane-associated area”

4.     Section 1.1. PSEN1 structure and functions: The authors state that “As part of the gamma secretase complex, PSEN1 and PSEN2 can interact with various proteins, including nicastrin (Nct), PSEN enhancer 2 (PEN2), and anterior pharynx 1 (APH-1).” I was somewhat uncomfortable with this sentence; a proper way could be, “As part of the γ-secretase complex, PSEN1 or PSEN2 forms a complex with nicastrin (Nct), PSEN enhancer 2 (PEN2), and anterior pharynx 1 (APH-1). Furthermore, the γ-secretase complex is known to interact with various other proteins, including…” (for instance, see binding proteins uncovered in Wakabayashi et al. 2009).

Wakabayashi T, Craessaerts K, Bammens L, Bentahir M, Borgions F, Herdewijn P, Staes A, Timmerman E, Vandekerckhove J, Rubinstein E, Boucheix C, Gevaert K, De Strooper B. Analysis of the gamma-secretase interactome and validation of its association with tetraspanin-enriched microdomains. Nat Cell Biol. 2009 Nov;11(11):1340-6.

Thank you, this issue has been fixed: “As part of the γ-secretase complex, PSEN1 or PSEN2 forms a complex with nicastrin (Nct), PSEN enhancer 2 (PEN2), and anterior pharynx 1 (APH-1). Furthermore, the γ-secretase complex is known to interact with various other proteins, including, ubiquitin, protein-folding related proteins or adhesion molecules [290].’

  1.     Section 1.1. PSEN1 structure and functions: The authors state “Knockout of the PSEN1 gene in mice resulted in elevated production of amyloid beta peptide (Ab42) due to abnormal processing of APP protein by gamma secretase, but it did not affect the cleavage of alpha and beta secretase.” This sentence is incorrect. De Strooper et al. (1998) showed in Fig. 3 that PSEN1 knockout results in a significant decrease in Abeta42 levels both in the medium and neuronal culture lysate. Furthermore, the authors state “A study by Yonemura et al. examined the gamma secretase activity of PSEN1 and PSEN2 in yeast models. PSEN1- gamma secretase represented significantly higher activity, compared to PSEN2 related gamma secretase in beta-galactosidase assay.” However, this study compared the activity of PSEN1 or PSEN2 per one γ-secretase complex and found that PSEN1/γ-secretase and PSEN2/γ-secretase exhibit similar activities. These apparent misunderstandings of previous works of literature raise a serious concern about whether this review article adequately delivers previous scientific findings to the readers.

Thank you, these issues have been fixed. “
PSEN1 is involved in the cleavage of APP C-terminal transmembrane region and the production of amyloid-
b peptide (Ab42) by g-secretase, after APP processing a-and b-secretases. PSEN1 may not be an enzyme itself, but a crucial regulator protein in g-secretase processing. PSEN1 could also play a role in the transport of C-terminal APP fragment to the gamma secretase complex. Deficiency in PSEN1 function may impair APP processing, resulting in altered amyloid peptide production”

“A study by Yonemura et al. examined the g-secretase activity of PSEN1 and PSEN2 in yeast models. PSEN1-g-secretase represented significantly higher activity, compared to PSEN2 related g-secretase in b-galactosidase assay. Amyloid peptide production resulted in higher amyloid production in case of PSEN1-g-secretase complex. However, this study also compared the PSEN1 and PSEN2 activity per one γ-secretase complex, and suggested that PSEN1 and PSEN2 may have similar degree of activity. Co-immunoprecipitation of PSEN1 and PSEN2 with other g-secretase proteins revealed that PSEN2 concentration was lower in the complex, compared PSEN1, suggesting that PSEN1 and PSEN2 may have different affinity to the other g-secretase component proteins”

  1.     Section 1.1. PSEN1 structure and functions: The authors state “The TM6 and TM7 domains contain two catalytic aspartates.”, which is correct; however, the citation [16] is incorrect. Wolfe et al., 1999, for the first time, reported that the two aspartates, a.a. number 257 and 385, are required for presenilin endoproteolysis and γ-secretase activity.

    Wolfe MS, Xia W, Ostaszewski BL, Diehl TS, Kimberly WT, Selkoe DJ. Two transmembrane aspartates in presenilin-1 required for presenilin endoproteolysis and gamma-secretase activity. Nature. 1999 Apr 8;398(6727):513-7.

Thank you, this reference has been replaced to the one you suggested.

Similarly, the reviewer found that there are many references not adequately cited. For example, [18] is incorrect, and I believe it should be De Strooper et al., 1999. This is an example, and the authors should fully ensure that the references they cited are correct.

De Strooper B, Annaert W, Cupers P, Saftig P, Craessaerts K, Mumm JS, Schroeter EH, Schrijvers V, Wolfe MS, Ray WJ, Goate A, Kopan R. A presenilin-1-dependent gamma-secretase-like protease mediates release of Notch intracellular domain. Nature. 1999 Apr 8;398(6727):518-22.

Thank you, this issue has been fixed.

  1.     Section 1.1. PSEN1 structure and functions: The authors state “Notch signaling has been verified to determine cellular fate and control cell differentiation and division.” This is correct; however, the authors should refer to citations. The reviewer also found many other sentences missing citation(s). For instance, the sentence “PSEN1 may also be involved in Wnt signaling by controlling β-catenin stability.” needs to cite Zhang et al., 1998. Again, this is only a few examples and I expect more; the authors need re-check of the entire manuscript to ensure if necessary references are adequately cited.

    Zhang Z, Hartmann H, Do VM, Abramowski D, Sturchler-Pierrat C, Staufenbiel M, Sommer B, van de Wetering M, Clevers H, Saftig P, De Strooper B, He X, Yankner BA. Destabilization of beta-catenin by mutations in presenilin-1 potentiates neuronal apoptosis. Nature. 1998 Oct 15;395(6703):698-702.

Thank you, this references and citation have been fixed.
Taken together, the reviewer highly recommends that the authors extensively revise the article by re-ensuring that 1) citation(s) is included when the authors describe scientific consensus, 2) the authors cite the original research articles, and importantly 3) the authors adequately summarize what previous original articles reported.

Thank you very much for the suggestions, we fixed the references, which may be incorrect. 

Reviewer 4 Report

The role of presenilins and their mutations is still controversial and functional investigations are complicated by the partial redundancy between PSEN1 and PSEN2 as well as their complex molecular biology. PSEN1 mutations are associated with earliest age of onset and atypical clinical features.

This manuscript is a good review of the genetics and functions of presenilin-1 gene, and gives an interesting perspective on the clinical phenotypes related to PSEN1 mutations, as well as on significant residues of the protein.

Minor points to be considered:

- It is well know that the mechanisms by which mutations in the presenilins genes cause familial Alzheimer disease are controversial. Although "gain-of-function" mechanism has been proposed, many PSEN1 mutations paradoxically impair gamma-secretase and ‘loss-of-function’ mechanisms have also been postulated.

It would be helpful to discuss better the mechanisms by which PSEN1 mutations might lead to neurodegeneration (amyloid vs presenilin hypothesis) and the dominant nature of the disease-associated mutations in the PSEN1 gene.

- Correct duplication "However, recent emerging studies on genome editing could reveal the pathological mechanisms of diseases, including AD AD [287]"

Author Response

The role of presenilins and their mutations is still controversial and functional investigations are complicated by the partial redundancy between PSEN1 and PSEN2 as well as their complex molecular biology. PSEN1 mutations are associated with earliest age of onset and atypical clinical features.

This manuscript is a good review of the genetics and functions of presenilin-1 gene, and gives an interesting perspective on the clinical phenotypes related to PSEN1 mutations, as well as on significant residues of the protein.

Thank you for the positive and constructive comments.

Minor points to be considered:

- It is well know that the mechanisms by which mutations in the presenilins genes cause familial Alzheimer disease are controversial. Although "gain-of-function" mechanism has been proposed, many PSEN1 mutations paradoxically impair gamma-secretase and ‘loss-of-function’ mechanisms have also been postulated. It would be helpful to discuss better the mechanisms by which PSEN1 mutations might lead to neurodegeneration (amyloid vs presenilin hypothesis) and the dominant nature of the disease-associated mutations in the PSEN1 gene.

Thank you, a chapter was added into the discussion

PSEN1 mutations could result in neurodegeneration through both gain-of function [PMID: 12130773] or loss-of-function mechanisms [PMID: 17197420]. Hardy and Selkoe (1992) proposed the amyloid hypothesis, suggesting that the accumulated amyloid peptides may be the main causative factors for Ad related neurodegeneration. Also, mutant PSEN1 mutations (such as Met84Val, Leu85Pro, His163Arg, His163Pro, Met233Leu) could enhance the APP processing and amyloid peptide (Ab42) generation. However, this study also made concerns about amyloid hypothesis. For example, amyloid plaques in the brain may not correlate with the degree of neurodegeneration. Also, amyloid production in cell cultures may not correlate with age of onset or disease phenotypes [PMID: 12130773]. Later, amyloid hypothesis was refuted further. It was revealed that elevated Ab42 may not be the only factor, leading to PSEN1-related neurodegeneration. For example, PSEN1 mutations may result in earlier disease onset, compared to mutations in APP, even though Ab42/40 ratio may not be as significant as expected [PMID: 24928124]. Also, amyloid overproduction did not result in significant neurodegeneration in mouse models [PMID: 7845465, PMID: 8810256, PMID: 10818140].  These findings suggested that loss of PSEN1 functions may play a crucial role in AD progression. PSEN knockout mice presented significant degree of neurodegeneration, however, the level of both Ab42 and Ab40 was reduced. PSEN knockout could result in elevated degree of neuroinflammation, reduced neuroprotection and increased degree of apoptosis [PMID: 15345711, PMID: 15066262, PMID: 20419112]. Shen and Kelleher (2007) proposed the presenilin hypothesis of AD, which may provide an alternative view of disease pathogenesis. Loss of essential PSEN1 (and PSEN2) functions may result in AD-related neurodegeneration. Several PSEN1 mutations (Gly209Arg, Gly209Val, Leu235Pro, Cys410Tyr, Leu435Phe) may not impact significantly the Ab42 levels (or even reduce it), but they may reduce significantly (or abrogate) the Ab40 production.  Pathogenic PSEN1 mutations may impair the g-secretase functions through dominant-negative mechanisms. Elevated amyloid levels may also inhibit further the g-secretase functions. Loss of PSEN activity may result in abnormalities in synaptic functions, leading neuronal loss, Tau hyperphosphorylation and dementia. Further studies are also needed on PSEN hypothesis. For example, APP mutations may not inhibit g-secretase related pathways. Also, not all PSEN1 mutations could result in Tau related pathology [PMID: 17197420, PMID: 29112196]. Besides amyloid and PSEN hypothesis, other AD-hypotheses were also described, such as Tau hypothesis, inflammation -hypothesis, cholinergic and oxidative stress hypothesis, confirming that AD is a very complex disease. There is no absolute hypothesis available on AD progression, but all possible hypotheses could provide explanation on AD progression and help the drug development [PMID: 19457065, PMID: 29112196, PMID: 29440986].

- Correct duplication "However, recent emerging studies on genome editing could reveal the pathological mechanisms of diseases, including AD AD [287]"

Thank you, issue has been fixed

Round 2

Reviewer 3 Report

The reviewer still found inadequate references, including [53], [67], [117], [160], [161], and [267].  

Author Response

The reviewer still found inadequate references, including [53], [67], [117], [160], [161], and [267]. 

Thank you very much for the suggestion, we tried to replace the references, or add new references to certain paragraphs.

Reference 53 was kept, but we added new references to the paragraph.

  • “Several mutations appeared in the TM1 region of PSEN1, the majority of which were confirmed to affect AD. The first TM domain contains several residues that could greatly impact the g-secretase cleavage. Residues in TM1 are highly conserved [64]. Furthermore, TM1 is located in proximity to critical motifs of g-secretase, such as GxGD and PALP, in TM7 and TM9, respectively. Mutations in TM1 may result in stress during the interactions between TM1 and TM7 (or TM9). Mutations in TM1 may also increase the distance between these helices [12, 53, 54,68, 263, 304].”
  • New references:
  • Gong, P.; Vetrivel, K.S.; Nguyen, P.D.; Meckler, X.; Cheng, H.; Kounnas, M.Z.; Wagner, S.L.; Parent, A.T.; Thinakaran, G. Mutation analysis of the presenilin 1 N-terminal domain reveals a broad spectrum of gamma-secretase activity toward amyloid precursor protein and other substrates. J. Biol. Chem. 2010, 285, 38042–38052.
  • Bagyinszky, E., et al., A Pathogenic Presenilin-1 Val96Phe Mutation from a Malaysian Family. Brain Sci, 2021. 11(10).
  • Szaruga, M.; Munteanu, B.; Lismont, S.; Veugelen, S.; Horré, K.; Mercken, M.; Saido, T.C.; Ryan, N.S.; De Vos, T.; Savvides, S.N.; et al. Alzheimer’s-Causing Mutations Shift Aβ Length by Destabilizing γ-Secretase-Aβn Interactions. Cell 2017, 170, 443–456.e14
  • Sato, C., et al., The C-terminal PAL motif and transmembrane domain 9 of presenilin 1 are involved in the formation of the catalytic pore of the gamma-secretase. J Neurosci, 2008. 28(24): p. 6264-71.
  • Sato, C., Morohashi, Y., Tomita, T., & Iwatsubo, T., Structure of the catalytic pore of gamma-secretase probed by the accessibility of substituted cysteines. The Journal of neuroscience : the official journal of the Society for Neuroscience,2006, 26(46), 12081–12088.

We replaced reference 67 with a new reference [306] to this paragraph:

  • “Mutant Glu120 in iPSC cells was associated with increased phosphorylated Tau levels and enhanced mitochondrial dysfunctions [306].”
  • New reference: [306]. Li, L., et al. iPSC Modeling of Presenilin1 Mutation in Alzheimer's Disease with Cerebellar Ataxia. Exp Neurobiol. 2018;27(5):350-364.

Reference 117 was kept, but at point, we agree, another reference [reference 126] may be more suitable

  • “In TM3, to date, 44 pathogenic or probable pathogenic mutations have been reported, suggesting that this region may be critical in gamma-secretase cleavage [126].
  • New reference: [126.] Senanarong, V., et al., Pathogenic PSEN1 Glu184Gly Mutation in a Family from Thailand with Probable Autosomal Dominant Early Onset Alzheimer's Disease. Diagnostics (Basel), 2020. 10(3).

Reference 160 has been replaced to reference 121 in this paragraph.

  • “Mutations in Met233 were related to the accumulation of APP beta-C-terminal fragments, which may impair the endosomal functions [121].”
  • New reference: [121]. Kwart, D., et al., A Large Panel of Isogenic APP and PSEN1 Mutant Human iPSC Neurons Reveals Shared Endosomal Abnormalities Mediated by APP beta-CTFs, Not Abeta. Neuron, 2019. 104(2): p. 256-270 e5.

Reference 160 was kept in this chapter, but an additional reference, ref161 was also added (see below)

  • “Leu381 may impair the carboxypeptidase-related g-secretase cleavage, leading to enhanced long amyloid production [160, 161, 227]

Reference 161  was changed     

  • New reference: [161]. Li, N., et al., Effect of Presenilin Mutations on APP Cleavage; Insights into the Pathogenesis of FAD. Front Aging Neurosci, 2016. 8: p. 51

Reference 267 was changed

  • New reference: [267]. Taddei, K., et al., Two novel presenilin-1 mutations (Ser169Leu and Pro436Gln) associated with very early onset Alzheimer's disease. Neuroreport. 1998;9(14):3335-3339